# Microglia: The Real Foe in HIV-1-Associated Neurocognitive Disorders?

**DOI:** 10.3390/biomedicines9080925

**Published:** 2021-07-30

**Authors:** Ana Borrajo López, Maria Aránzazu Penedo, Tania Rivera-Baltanas, Daniel Pérez-Rodríguez, David Alonso-Crespo, Carlos Fernández-Pereira, José Manuel Olivares, Roberto Carlos Agís-Balboa

**Affiliations:** 1Department of Microbiology and Parasitology, Faculty of Pharmacy, Complutense University of Madrid, 28040 Madrid, Spain; 2Department of Experimental Medicine and Surgery, University of Rome Tor Vergata, 00133 Roma, Italy; 3Translational Neuroscience Group-CIBERSAM, Galicia Sur Health Research Institute (IIS Galicia Sur), Área Sanitaria de Vigo-Hospital Álvaro Cunqueiro, SERGAS-UVIGO, 36213 Vigo, Spain; aranchapenedo@gmail.com (M.A.P.); tania_baltanas@hotmail.com (T.R.-B.); daniel_zim93@hotmail.com (D.P.-R.); carlosfernandezpereira@gmail.com (C.F.-P.); jose.manuel.olivares@gmail.com (J.M.O.); 4Neuro Epigenetics Laboratory, University Hospital Complex of Vigo, SERGAS-UVIGO, 36213 Virgo, Spain; 5Nursing Team-Intensive Care Unit, Área Sanitaria de Vigo, Estrada de Clara Campoamor 341, SERGAS-UVigo, 36312 Virgo, Spain; davidalcrep@gmail.com; 6Department of Psychiatry, Área Sanitaria de Vigo, Estrada de Clara Campoamor 341, SERGAS-UVigo, 36312 Vigo, Spain

**Keywords:** HIV-1 associated neurocognitive disorders, microglia, pharmacological treatment, central nervous system, antiretroviral therapy

## Abstract

The current use of combined antiretroviral therapy (cART) is leading to a significant decrease in deaths and comorbidities associated with human immunodeficiency virus type 1 (HIV-1) infection. Nonetheless, none of these therapies can extinguish the virus from the long-lived cellular reservoir, including microglia, thereby representing an important obstacle to curing HIV. Microglia are the foremost cells infected by HIV-1 in the central nervous system (CNS) and are believed to be involved in the development of HIV-1-associated neurocognitive disorder (HAND). At present, the pathological mechanisms contributing to HAND remain unclear, but evidence suggests that removing these infected cells from the brain, as well as obtaining a better understanding of the specific molecular mechanisms of HIV-1 latency in these cells, should help in the design of new strategies to prevent HAND and achieve a cure for these diseases. The goal of this review was to study the current state of knowledge of the neuropathology and research models of HAND containing virus susceptible target cells (microglial cells) and potential pharmacological treatment approaches under investigation.

## 1. Introduction

Human immunodeficiency virus 1 (HIV-1) establishes a chronic infection which can lead to severe immunodeficiency despite treatment with potent combined antiretroviral therapy (cART). HIV-1 can evade immune responses through several mechanisms, including the establishment of persistent infection within different cell types such as the microglia and alveolar macrophages [1,2]. Macrophages (including those residing in the brain, known as microglia) can travel to different anatomical sites, thus allowing HIV-1 to spread through the body of an infected person [2,3,4,5]. The brain is a typical example of an anatomical compartment with poor drug penetration and reduced immune surveillance because of the blood brain barrier (BBB) [6] (Figure 1). Moreover, the BBB represents a potential factor limiting the efficacy of combined antiretroviral therapy (cART) in the central nervous system (CNS).

Early rebounding of plasma HIV-1 is not explained by the pool of resting CD4^+^ T cells. Past phylogenetic work has shown that HIV-1 production in the brain is associated with the emergence of virus resistance [7,8] and with HIV-1-associated neurocognitive disorder (HAND) [9]. HAND comprises three disorders, each associated with increasing levels of morbidity and mortality: asymptomatic neurocognitive impairment (ANI), mild neurocognitive disorder (MND), and HIV-associated dementia (HAD) [10].

HAD is the most serious form of HAND and is described as a severe dementia that manifests as a concentration deficit, notable motor problems, and fluctuating behavioral alterations [10]. Efficient viral suppression with cART has now improved the prognosis of HAND from HAD to that expected from ANI or MND [10]. MND causes mild interference in the actions of daily living and can be determined through self-reported decreased mental acuity or observations by people close to the patient [10]. A good diagnosis of MND includes patients that have an acquired cognitive function impairment in two of the listed domains with scores greater than one standard deviation below the demographically corrected means [10]. ANI is the most frequent form of HAND and involves two cognitive domains that do not interfere with everyday function [6].

HAND still represents an unresolved multifactorial complication of HIV, even in the modern cART (traditional three-drug regimen) era that often achieves the suppression of HIV replication to levels below the limit of detection of currently used assays. Indeed, recent studies have demonstrated that no clinical trials of HAND treatments are efficient beyond the optimal suppression of HIV replication in the CNS [11,12]. Whereas some studies have indicated a reduction in cognitive impairment due to cART, patients with a controlled viral load who discontinue treatment with antiretrovirals have shown an improvement in cognitive functions and a reduction in neuronal damage [12,13].

Another crucial strategy is based on the detection of latent viral reservoirs, although these methods have not yet been studied in depth. They are also limited because the degree of HIV elimination cannot be estimated, and latency reversing agents (LRA) might pose a risk of inflammatory over-response by activating latent viral reservoirs in the brain in immunocompetent patients, thereby risking further neurological damage [14].

Despite the critical relevance of microglia infection in HAND, studies that characterize the replication of HIV-1 in these cells are scarce as they involve using primary cultures obtained from fetal brains or the post-mortem assessment of AIDS patients, which are both difficult to achieve [14]. Therefore, the mechanism of HIV-1 action in microglia remains largely unknown. The pathogenesis of HAND is still unknown but appears to be caused by multiple factors: cART neurotoxicity, replication of HIV in the CNS from infected latent cells, inflammation of the CNS, calcium dysregulation, mitochondrial dysfunction, drug abuse, and autophagy [6].

Furthermore, the generation of newly synthesized viral proteins, such as trans-activator of transcription (Tat) or viral surface glycoprotein 120 (gp120) from infected cells, might lead to direct neural injury [6,8]. In fact, recent studies have shown new mechanisms for HIV-1 Tat-mediated microglial inflammation which involve a novel micro ribonucleic acid (miRNA)-34a- NOD-like receptor C5 (NLRC5)-NFκB signaling axis or activation of the methyl CpG binding protein 2 (MEPC2)-STAT3 axis [15]. Activation by Tat of the microglial nucleotide-binding domain leucin-rich repeat and pyrin-containing receptor 3 (NLRP3) inflammasome can also provoke neuroinflammation [8,16] (Table 1).

Of note, microglia express receptors for various neurotransmitters as well as for innate immunity ligands, including pattern-recognition receptors such as toll-like receptors [6]. Furthermore, microglial cells present antigens and secrete cytokines [17]—low molecular weight proteins generally classified as pro- or anti-inflammatory types—involved in the physiological processes implemented to combat pathogens and repair tissues. Whereas proinflammatory cytokines can elicit a sustained immune response, anti-inflammatory cytokines downregulate immune responses by binding to receptors expressed in the microglia to initiate an autocrine signaling process. Cytokines have several effects on the function of the CNS, such as promoting the growth and proliferation of astrocytes and oligodendrocytes and modulating the release of neurotransmitters [6]. Several studies have indicated that low levels of proinflammatory cytokines are expressed in the healthy brain. However, their expression is high in brain and cerebrospinal fluid (CSF) samples from patients with Parkinson’s disease, Alzheimer’s disease [6], and other psychiatric diseases such as schizophrenia [6].

Macrophages or microglia can provoke the release of pr-inflammatory cytokines, including tumor necrosis factor α (TNFα), interferon α (IFNα), interleukin-6 (IL6), interleukin-8 (IL8), and interleukin-1β (IL1β) [6,8,13] (Figure 1), as well as chemokines (C-C motif chemokine ligand 2 (CCL2) and C-C motif chemokine ligand 5 (CCL5), macrophage inflammatory protein 1β (MIP)-1β) [17]. These clues indicate the existence of cellular reservoirs in the CNS established within 3 to 5 days of HIV-1 infection [8], comprising three different types of long-lived infected cells [18]: astrocytes, monocyte lineage cells [19], and microglial cells [20].

One important study showed the criteria for the formation of a cell reservoir: (i) the presence of HIV-1 deoxyribonucleic acid (DNA) integrated into the host genome of long-lived cells; (ii) the existence of mechanisms which permit the virus to persist for long periods in latent cells, including mechanisms allowing the establishment and maintenance of a latent infection; and (iii) the formation of replication-competent particles following activation of these reservoirs [21]. These long-lived infected cells satisfy two of these criteria: the presence of HIV-1 DNA integrated into the host genome of long-lived cells, and the existence of mechanisms allowing the virus to persist for prolonged period in latent cells [8].

Nonetheless, different studies using the most sensitive detection methods have shown that, although astrocytes are the most abundant infected cell type in the CNS [22], HIV-1 is not detected in these cells but is found in macrophages and microglial cells [23]. Furthermore, astrocyte infection seems to be nonproductive. Thus, taken together, this evidence suggests that these cells are not real viral reservoirs for HIV-1 [8].

However, a recent study suggested that microglial cells are highly susceptible to HIV-1 infection and provoke productive infections [24] correlated with HAND in humans and in macaque animal models. Thus, these long-lived cells may be HIV reservoirs in the CNS. Nonetheless, there are differences in the size and constitution of these cells in the brain, meaning that these two cell types are not equally relevant. Macrophages are bone marrow-derived with a half-life of months and do not undergo cell division [25]. Instead, they are repopulated from monocytes crossing the BBB.

The use of effective cART or other therapeutic strategies prevent the specific subset of infected monocytes from crossing the BBB, which can lead to drastic decreases of these cells in the brain [8]. In contrast, microglial cells originate from erythromyeloid progenitors in the yolk sac during embryogenesis and colonize the forming CNS [26]. These cells constitute the main residents of the brain, and because of their long half-life (in years), they form a stable population [27]. Moreover, unlike macrophages, they can undergo cell division, enabling HIV-1 to persist in the brain [8,28]. Thus, it seems likely that microglial cells might constitute one of the principal HIV-1 reservoirs in the brain.

In summary, infection of long-lived cells such as microglia in the CNS is believed to contribute not only to the progression of HAND, but also serve as a long-term reservoir, resulting in the persistence of HIV-1 in a sanctuary site, which is important for the treatment and potential eradication of infection. In this review, we discuss the establishment and maintenance of HIV-1 latency in microglial cells and the therapeutic implications of these mechanisms. We then discuss the novel potential treatment approaches being investigated for HAND.

## 2. Microglia Biology

Microglia are the resident immune cell population of the CNS and were discovered by Franz Nissl, Alois Alzheimer, Ramón y Cajal, and Pío del Río-Hortega during the 19th and 20th centuries [29]. Microglia, which can migrate, proliferate, and phagocytose, constitute 10–15% of all CNS cells, and can be differentiated from neurons and other types of glial cells present in the brain by their morphological characteristics [26]. They establish numerous contacts with other CNS cells, including neurons, astrocytes, and oligodendrocytes.

Early studies indicated that, similar to neurons and glial cells, microglia had a neuroectodermal origin. However, recent work has demonstrated that microglial cells are macrophages that belong to the immune system. Crucial studies in mice revealed that macrophages can be derived from different developmental pathways that differentially contribute to the respective tissue compartments in embryos and adults [30]. These cells come from myeloid precursors with ‘primitive’ and ‘definitive’ erythroid myeloid potential which are consumed before birth, as well as from hematopoietic stem cells that establish and maintain definitive adult lymphoid and myeloid hematopoiesis throughout adulthood [31]. Microglial cells originate from c-Kit^lo^CD41^lo^ progenitors in the embryonic yolk sac early during embryogenesis [31] and have the potential for primitive erythropoiesis.

Microglia cells were first identified and ultrastructurally characterized in the corpus callosum in the mature brain [31]. Most of microglial cells in adult brain tissue are in a ‘resting’ (quiescent) state and typically have a branched morphology. In 1996, Kreutzberg et al. [32] used the in vivo morphology of microglia to classify them as (a) branched with small cells and many thin branches (quiescent), or (b) ameboid with truncated processes (an active status) to simplify proliferation, migration, and phagocytosis [33]. Nonetheless, quiescent microglia are not static, rather, their branches constantly move. Moreover, work in mice has demonstrated that they dynamically expand and retract [34,35].

The transformation of amoeboid microglia into ramified microglia is a process that coincides with the onset of axon myelination in the corpus callosum [31]. Electron microscopic studies have shown that amoeboid cells in 1- to 5-day-old animals possess a round nucleus with marginal chromatin clumps and abundant cytoplasm displaying lysosomal dense granules, vacuoles, and a well-developed Golgi apparatus. In contrast, most microglia in older animals are elongated and branched, and have a flattened nucleus and scant cytoplasm containing only a few lysosomal granules [31]. These cells are immunoreactive with the microglia specific antibody OX42 and can also be labelled with lectin.

Accumulating evidence in recent years supports the idea that microglia are immune-regulatory cells that play important roles in the CNS in health and disease. They help maintain the homeostasis of the brain environment under normal conditions but also produce robust reactions in response to inflammatory stimuli, injury, hypoxia-ischemia, and other adverse conditions affecting the normal function of the brain. Activated microglia take on an amoeboid morphology, proliferate, and migrate to the site of injury or damage where they protect the CNS from viruses and pathogens [31]. However, overactivation and/or chronic activation of microglia with the excess production of inflammatory mediators may result in neurotoxic consequences [36]. Indeed, microglial cells can contribute to neurocognitive degeneration, as seen in the different forms of HAND, leading to an increase in proinflammatory chemokines and cytokines, as well as neurotoxins, which can affect both astrocytes and neurons and can cause neuronal damage [37].

## 3. Microglia as Main HIV-1 Reservoir in the CNS

Microglial cells play essential roles in the clearance of amyloid-beat (Aβ) and tau proteins [38] in the context of Alzheimer’s and Parkinson’s disease, respectively. The accumulation of these proteins in these neurodegenerative diseases significantly correlates with the presentation of neurocognitive impairments. Novel transcriptomics studies have shown that homeostatic microglia can adopt a phagocytic disease-associated microglia (DAM) phenotype in the context of neurodegenerative disease, chronic neuroinflammatory states, and advanced aging [39]. Network-based meta-analysis procedures applied to microglial transcriptomes have suggested that microglial cells take on a wide range of functions in neurodegeneration, and microglial transcriptomic data have indicated increased phago-lysosome, oxidative phosphorylation, and antigen-presentation functions in these cells [39].

The myeloid protein, Trem 2, which participates in microglial survival and proliferation, controls a checkpoint required for the DAM conformation. Removal of Trem2 causes increased neuritic injury and prevents microglial accumulation around Aβ plaques [40]. These data indicate that DAM may promote more efficient phagocytosis, and that removal of the pathological protein aggregates present in neurodegenerative disorders. These findings also highlight the great complexity of microglial functional roles in neurodegeneration and indicate that DAM have different proinflammatory functional states. A novel bulk transcriptome analysis applied to re-analyze microglial single-cell ribonucleic acid (RNA) sequence data described the presence of diverse microglial co-expression modules [40]. The analysis also determined novel interferon-related and lipopolysaccharide (LPS)-related co-expression networks, and so-far unknown microglial sub-populations, in addition to DAM in neurodegeneration models [40].

Another critical study showed functional changes in microglia during HIV-1 infection and aging, including cellular activation, arrested cell cycle progression, and altered cellular metabolism [40,41]. Some proinflammatory cytokines belong to the unique secretory cellular senescence phenotype, which is known to occur during aging and which may also be the origin of tissue and organismal aging. Moreover, microglia also express higher protein levels of cell cycle inhibitors and exhibit altered autophagy, both of which are integral parts of cellular senescence [42,43]. Furthermore, they are principal players in pathological neuroinflammation events observed in patients receiving cART.

As described above, HAND is thought to be associated with the release of excessive cytokines and chemokines [44,45]. Ginsberg et al. (2018) demonstrated that the gene expression profiles in microglial cells from infected patients receiving cART with or without HIV encephalitis (HIVE) were different compared to those of non-infected patients. In total, 64% of the genes were affected in HIVE-positive patients, while only 24% were involved in HIVE-negative patients [44]. Thus, this study suggests that neuroinflammation, which leads to neurodegenerative diseases such as HAND, is related to microglial damage.

Neuroimaging using techniques such as positron emission tomography (PET) has revealed that neuroinflammation develops even in patients effectively treated with cART [8,46,47], thereby highlighting the importance of microglial activation in HIV+ patients with efficient cART treatments, whether or not they have also suffered cognitive damage. PET can also be used to follow the level of microglial activation by targeting the translocator protein (TSPO) [48]. Microglia activation is associated with the progression of neurodegenerative diseases, impacting the BBB [8,49] by modulating BBB endothelial cell tight junctions [50] (Figure 1). In contrast, inhibition of microglial activation by minocycline is associated with preservation of the BBB integrity in vitro and with reduced BBB disruption in vivo [51].

In conclusion, microglial action is linked to the development of several neurological disorders, and this impairment plays a crucial role in the pathogenesis of HAND by contributing to neurodegenerative events via different mechanisms. Moreover, because these cells are resistant to the cytopathic effects of HIV-1, they can sustain the infection for prolonged periods of time [52]. They subsequently become involved in inflammation by releasing HIV-1 proteins, inflammatory cytokines, and neurotoxins, which then induce astrocyte differentiation and apoptosis, and alter normal neurogenesis [52] (Figure 1). Thus, microglia play a crucial role in mediating neurodegeneration processes. Despite this, it is important to note that immunological response in the CNS does not depend only on microglial cells, and that cytokines bind receptors other than those on microglia. Thus, HAND is also the consequence of the immunological response of other types of neural cells.

## 4. The Entry of HIV-1 into Microglial Cells and Their Susceptibility to the Virus. Relevant Studies about the Mechanism of Infection and Molecular Mechanisms Involved in Establishing and Maintaining HIV-1 Latency in Microglia, and Procedures to Study HIV Infection in Microglia

It is well established in rodents, non-human primates, and humans that monocyte/macrophage trafficking to the CNS is normal both in health and disease [53]. When HIV and simian immunodeficiency virus (SIV) infections occur, the number of activated monocytes and turnover of monocyte/macrophages in tissues increases [53]. Rappaport et al. found a population of expanded CD16^+^CD163^+^ monocytes in blood that may replace or take up residence and accumulate in the CNS of HIV-infected individuals [52]. As previously mentioned, transmigration of this macrophage population—which is highly susceptible to HIV infection—across the BBB into the CNS is critical to the pathogenesis of HAND. Work analyzing some of the mechanisms contributing to the entry of this mature monocyte subset into the brain has shown the presence of extremely increased sensitivity to CCL2 upon CC-chemokine receptor 2 (CCR2)-mediated HIV infection [54,55].

Not only is the presence of CD4 required for HIV-1 to enter a CNS cell, but the CXCR4 or CC-chemokine receptor 5 (CCR5) co-receptors and CC-chemokine receptor 3 (CCR3) receptors are also required to produce effective infection. Of note, CCR5 is more strongly associated with virus entry and the subsequent development of neurological diseases. Microglia express both CCR3 and CCR5 on their surface [56]. Thus, they are more susceptible to HIV infection [56] (Figure 1).

Different latency mechanisms in microglial cells have also been described. An important study in a microglial cell line showed that the high mobility group AT-hook 1 (HMGA1) protein is recruited to an inactive complex formed by the COUP transcription factor (COUP-TF) interacting protein 2 (CTIP-2) and positive transcription elongation factor (P-TEFb) on the HIV-1 long terminal repeat (LTR) [57]. Geffin et al. used human fetal microglia stimulated with interferon (IFN)-α and infected with HIV to show that HIV was unable to generate viral RNA and DNA and that viral spread was limited because of the induction of tetherin expression, a host restriction factor [36] (Table 1).

Recent findings suggest that HIV-1 infection can produce an inflammatory environment due to the induction of viral proteins (such as Tat and gp120) and proinflammatory cytokines (e.g., TNF-α, IL-8, IL-6, and IL-1β) [72,73]. The HIV-1 Tat protein can activate the NLR and increase caspase-1 and IL-1β levels in microglia, which, in turn, generates TNF-α and IL-6 and intensifies the proinflammation response [16] (Table 1). HIV-1 infection in microglial cells can also produce reactive oxygen species (ROS) and reactive nitrogen species (RNS), such as quinolinic acid, arachidonic acid, and nitric oxide [55]. Moreover, the presence of HIV-1 in the brain can cause the expression of inducible hypoxia factor (HIF)-1, leading to the generation of mitochondrial dysfunction and oxidative stress [52,74]. Other recent work also suggests that HIV-1 infection in primary human microglia cells causes the release of IL-6, IL-8, TNF-α, CCL2, and regulated-upon-activation, normal T cell-expressed and -secreted (RANTES) proteins [59]. This latter study demonstrated that HIV-1 infection induced autophagy, which was correlated with beclin 1, in turn activating the release of the p24 viral protein and secretion of proinflammatory cytokines. Thus, replication of HIV-1 in microglial cells and their activation are dependent on the activation of autophagy [57].

Despite the pivotal role of microglial cells in HAND, few studies have focused on HIV-1 replication in microglia and the different mechanisms of the virus action. Some studies have used primary cultures from fetal brains or have performed post-mortem assessments of acquired immune deficiency syndrome (AIDS) patients, which is difficult to access in both cases [72,73] (Table 1). A more recent study developed a stable immortalized human microglial cell line. The cell line has a microglia-like morphology, expresses key microglial surface markers, has appropriate migratory and phagocytic activity, has the capacity to establish an inflammatory response characteristic of primary microglia, and could be latently infected with HIV-1 proviruses. Thus, this study provides a well-characterized cell line that represents a helpful instrument for studying microglial cell function and the mechanics and dynamics of HIV-1 replication in the CNS [59,74] (Table 1).

Other work has examined the mechanisms of HIV-1 infection. In 2017, Mlcochova et al. showed that the absence of restriction by sterile alpha motif and histidine/aspartic acid domain-containing protein 1 (SAMDH1) restriction factor (RF) sterile alpha motif (SAM) was because it was phosphorylated by cyclin kinase 1 (CDK1), and it itself was induced in microglial cells cycling between G_0_ to G_1_ states [61] (Table 1). Taken together, these clues indicate that microglia are infected by HIV-1 both in vitro and in vivo [20].

Another study assessing SAMHD1 restriction in in vitro differentiated macrophages and freshly isolated monocyte-derived macrophages from lung (alveolar), abdomen (peritoneal), and brain tissue found that infection and spread in in vitro cultured macrophages was very limited and that the action of HIV Vpx protein was largely restricted to the initial infection. Nearly identical infection and restriction profiles were observed in freshly isolated peripheral blood monocytes, as well as lung and abdominal macrophages [61]. In contrast, under the same infection conditions, microglia were highly susceptible to HIV-1 infection, despite the levels of endogenous SAMHD1, compared to other macrophage populations. Addition of Vpx further enhanced HIV-1 infection in a virus input restricted environment, and viral spread was robust regardless of SAMHD1 consumption [61]. These results demonstrate that HIV-1 infection of peripherally circulating macrophages is effectively limited by SAMHD1, but that microglia are highly susceptible to infection in spite of SAMHD1 expression. This work explains the long-standing observation that HIV-1 infection is often detected in microglia in the brain, but seldom in other tissues of the body [61] (Table 1).

Cosenza et al. and Churchill et al. identified HIV-1 DNA, RNA, and proteins in microglial cells from HIV-patient autopsy tissues, although we must emphasize that these patients died from severe HAND [62,63] (Table 1). The presence of HIV DNA in microglial cells and macrophages was also detected in brain autopsies of patients with controlled HIV-1 infections [64,75] (Table 1). A novel study showed that microglia were infected in patients with suppressed viral levels who died from causes not related to HIV-1 [23]. This work used a unique cohort from the National Neuro AIDS Tissue Consortium (NTTC) comprising 16 patients on cART with well-documented, sustained HIV-1 control. The authors utilized highly specific technology to detect and quantify both HIV-1 DNA and RNAs at the cellular level, and the results showed that both perivascular macrophages and microglia, but not astrocytes, harbored HIV-1 DNA. In 6 out of 16 cases, the authors also found HIV-1 RNA in these cells when HIV-1 RNA was undetectable in cerebrospinal fluid (CSF) and blood, thus demonstrating that virus generation can take place in the CNS.

Several other studies have also demonstrated that microglia are susceptible to infection in in vitro in models of human microglial cells [56,76]. A few latency models have been derived from these earlier models and have become important tools for studying the mechanism of infection and molecular mechanisms underlying the establishment and maintenance of HIV-1 latency in microglia [56,77] (Table 1). Confirmation of virus detection in CSF from patients undergoing effective cART who otherwise had undetectable plasma HIV-1 levels highlights the fact that HIV-1 can be produced in the brain [60] (Table 1). In addition, phylogenetic analyses have suggested that HIV-1 can become compartmentalized in the CSF even early on during infection [65,78] (Table 1).

Other important work has studied the action of microglia in white matter along with the multinucleated giant cells that are prominent features of HIVE and of several SIV models [66]. Whereas infected microglial cells have been demonstrated in HIVE, this characteristic is not usually seen in experimental infections in rhesus macaques using SIV or chimeric simian/HIV (SHIV) strains, thus limiting their utility in HIV-1 pathogenesis and treatment studies. Importantly, this study showed infection of microglial cells, which makes R5 SHIV_SF162P3N_ infection of macaques an interesting animal model not only to study transmission and HIVE pathogenesis but also to conduct preclinical evaluations of therapeutic interventions aimed at eradicating HIV-1 from the CNS [66] (Table 1). Several other studies have demonstrated microglial cell infections in animal models, such as in macaques and humanized mice [67,68,79,80], where HIV-1 DNA, RNA, and protein were detected in the brains of these animal models.

There are also remarkably interesting studies of latent infection of the microglia [81]. For example, a mechanism for the establishment of HIV-1 transcription was proposed in a macaque model [82] in which microglia could be reactivated in response to cytokine stimulation [82,83]. Moreover, other important studies have shown that the glucocorticoid receptor is a critical repressor of HIV transcription in microglia and a novel potential pharmacological target to restrict HIV expression in the CNS (reviewed in [77]). Other studies carried out in the same laboratory have demonstrated that TLR receptor-mediated responses to inflammatory conditions by microglia could provoke the induction of latent HIV proviruses and contribute to the etiology of HAND [84]. Furthermore, humanized mouse models were produced in which microglia were infected by HIV-1 in vivo [58,69,85,86] (Table 1). These models allow us to study the pathophysiology of microglial cell activation and to develop strategies aimed at reducing the pool of these reservoirs.

In summary, there is significant evidence that microglia constitute an important cellular reservoir for HIV in the brain. The possible drastic reduction of this reservoir is also being studied by intensifying cART or preventing the transmigration of infected monocytes to the brain, but it is not yet clear whether astrocytes constitute a true reservoir [70,71] (Table 1). There is a lack of knowledge of the specific molecular mechanisms underlying the persistence of HIV-1 latency in microglial cells. Therefore, future studies that focus on these issues will be required to design original strategies aimed at targeting these reservoirs.

## 5. General and Basic Mechanisms of Latency

In latently infected cells, most proviruses are found in actively transcribed genes [87]. Certain mechanisms can contribute to silencing HIV gene expression and replication, including deleterious mutations in the viral genome (several of which can be repaired by recombination if more than one virus integrates into the same cell) [88], transcriptional interference [89], changes in chromatin structure [90], epigenetic silencing (such as increased DNA methylation) [91], the presence of negative and the absence of positive transcription factors (TFs) [92], and problems with RNA processing and transport [93].

### 5.1. Pre-Integration Latency

Integration of HIV-1 pro-viral DNA into the host genome is an indispensable feature for favorable viral pathogenesis. Upon HIV-1 entry into the cell, HIV-1 RNA is reverse transcribed into DNA and further assembled in the form of the pre-integration complex (PIC) [93]. The PIC is composed of the viral proteins integrase, matrix, capsid, viral protein R (Vpr), and dsDNA. The PIC is later transported into the nucleus where it may integrate into the host genome. Pre-integration latency occurs because of poor reverse transcription effectiveness and restriction of PIC nuclear transportation [52] (Figure 2). Data have shown that pre-integration latency is quite common in vivo and, moreover, it rapidly decays in resting T cells [94]. In contrast, limitation of the dNTP pool and the presence of several host RFs control HIV-1 replication in monocyte- and macrophage-lineage cells and may contribute to pre-integration latency. These RFs are apolipoprotein B mRNA editing enzyme catalytic polypeptide-like (APOBEC)3 [95], the SAM domain, SAMHD1 [96], and myxovirus resistance 2 (MX2), which is also referred to as MXB [97]. APOBEC3 plays a pivotal role in causing G to A hypermutation of the HIV-1 genome (Figure 2). In particular, SAMHD1 reduces the pool of dNTPs present in macrophages by hydrolyzing them into their precursors (nucleosides and triphosphates), thereby resulting in ineffective viral reverse transcription [96] (Figure 2). Furthermore, MX2 restricts HIV replication in several susceptible cell types including macrophages and microglia [97]. Moreover, MX2 blocks HIV infection at the post-entry level by hindering the nuclear accumulation and integration of pro-viral DNA into host chromatin [97].

One important study screened 19,121 human genes in HIV-1 infected cells using ana siRNA and library, and host restriction factors identified 114 genes that significantly influence HIV infection [98]. In addition, blocking all the members of the RNA polymerase II-associated protein 1 (PAF1) family expressed in monocyte and macrophage-lineage cells [96] enhanced the effectiveness of HIV reverse transcription and pro-viral DNA integration [98]. Nevertheless, HIV-1 accessory viral proteins such as Vif [99] and Vpx [100] have been observed to target these restriction factors in monocyte- and macrophage-lineage cells. Vif impedes the synthesis of APOBEC3 mRNA, and Vif and Vpx induce the degradation of APOBEC3 protein via the 26S proteasome [52,99].

### 5.2. Post-Integration Latency

The mechanism of post-integration latency has been well established, with the integration of HIV-1 pro-viral DNA into host chromatin followed by silencing of HIV-1 gene expression, which is the real reason for the latent persistence and dissemination of HIV-1 (Figure 3). Some potential mechanisms, such as epigenetic gene silencing [101], transcription gene silencing (TGS), and post transcriptional gene silencing, have been shown to establish and maintain latency in target cells [102].

#### 5.2.1. Transcriptional Interference

The most probable cause of this lack of HIV gene expression is transcriptional interference. The 5′ LTR of HIV-1 initiates transcription 20-times more often than the 3′ LTR (which terminates transcription), and this is blocked by the elongating RNA polymerase II (RNAPII; Figure 3) [91]. Transcriptional interference is partly due to low-affinity binding between the DNA at the 3′ LTR and specificity protein 1 (SP1), TFs, and the initiator element [92]. Elongating RNAPII from the 5′ LTR moves these TFs from DNA at the 3′ LTR, then progresses to the polyA site in the R region and finishes transcription.

Thus, when the virus integrates in the same orientation as the host gene, the elongating RNAPII from the host gene finishes at the polyA site in the 5′ LTR, curtailing HIV transcription [92]. In this scenario, the 3′ LTR is unblocked and initiates transcription, causing usable transcripts, including the transactivation response (TAR). However, when the virus integrates in the opposite orientation, RNAPII copies HIV antisense transcripts, ignoring both polyA sites in the HIV sense orientation and generating a long hybrid mRNA species (Figure 3). If the provirus integrates into introns of host genes, then the HIV antisense transcripts are spliced out and are rapidly degraded. However, HIV antisense transcripts have also been discovered [103], and these could provide an estimate of transcriptional interference in latently infected cells.

#### 5.2.2. Host Integration, Heterochromatin, and Epigenetic Alterations

The host chromatin comprises heterochromatin (densely packed and transcriptionally inactive) and euchromatin (loosely packed and transcriptionally active) regions [104]. Insertion of HIV-1 into the heterochromatin regions can favor latency. However, findings suggest that HIV-1 pro-viral DNA preferentially integrates into euchromatin regions [105]. Of note, several studies have indicated that histone H3 lysine 9 trimethylation (H3K9me3) plays a role in heterochromatin formation and transcriptional silencing of integrated HIV-1 [106].

Heterochromatin can silence integrated proviruses in actively transcribed genes, gene deserts, and Alu-rich repeats. Therefore, increased levels of DNA methylation or chromatin silencing complexes, decreased overall cellular gene transcription, or insufficient positive TFs can help to recruitment nucleosomes to the viral genome and the HIV-1 LTR [102]. Moreover, nucleosomes situated at the U5 and R regions block the movement of RNAPII and thus inhibit HIV-1 replication [102]. This silencing results from the absence of crucial TFs (whose levels are gradually reduced as activated lymphocytes transition to memory T cells) and can be overcome by activating cells and increasing Tat synthesis [107].

#### 5.2.3. Involvement of Crucial Host Transcription Factors and Viral Proteins in Latency

As we previously mentioned, integrated HIV-1 pro-viral DNA is delimited at both ends by the LTR. The 5′LTR has unique blend of a robust TATA box, a potent initiator sequence, and binding sites for several TFs, including the glucocorticoid receptor COUP, USF, activator protein 1 (AP1), TCF-1α, C-Myc, SP1, CTF/NFE, NFAT, NF-κβ, and Tat [52]. In addition, two nucleosomes (nuc-0 and nuc-1) are precisely positioned on the HIV-1 LTR in latently infected cells [52]. Nuc-1 is positioned immediately downstream of a transcription start site and blocks transcription initiation and elongation (Figure 3). These TFs can be crucial for pro-viral DNA expression.

In addition to cellular TFs, viral latency can also be influenced by Tat. HIV-1 transcription can take place even in the absence of Tat but can only generate prematurely terminated short transcripts [108]. P-TEFb helps the production of a complete transcript from host or pro-viral DNA [107] and consists of a regulatory subunit, a large cyclin CycT1, and catalytic subunit—a cyclin-dependent kinase 9 (CDK9) [109]. In the absence of Tat, RNAPII is delayed after the transcription of TAR. Tat recruits P-TEFb via CycT1 to TAR, allowing CDK9 to phosphorylate the C-terminal domain (CTD) of RNAPII. In addition, CDK2 phosphorylates Serine 90 (Ser90) on CDK9 to also assist in HIV-1 transcription [109] (Figure 3).

The RNAPII CTD includes 52 YSPTSPS heptapeptide repeats in which all the serines, the tyrosine, and the threonine can be phosphorylated by different kinases. These modifications, especially the phosphorylation of serines at position 2 (Ser2P) and 5 (Ser5P), play pivotal roles in the co-transcriptional processing of nascent mRNA species [110]. In addition, CycH/cyclin-dependent kinase 7 (CDK7) phosphorylates Ser5, which is required for capping of all viral transcripts [87,109]. Notably, P-TEFb phosphorylates Ser2, thereby promoting efficient elongation of viral transcription [87].

P-TEFb also regulates the activity of factors that impair transcript elongation. P-TEFb phosphorylates the transcription elongation factor SPT5, which is part of the 5,6-dichloro-1-β-d-ribofuranosylbenzimidazole (DRB) sensitivity-inducing factor (SIF) complex and the (RNA extracted from RD-114 virions) RD subunit of the negative elongation factor (NELF) complex (NELF-E). These two complexes arrest RNAPII at TAR [111]. When (transcription elongation factor SPT5) SPT5 is phosphorylated, DSIF is transformed to a positive elongation factor. When NELF-E is phosphorylated, NELF disengages from TAR so that RNAPII can transition from initiation to elongation [111]. Thus, when P-TEFb levels are low, HIV is not transcribed. P-TEFb is also the NF-κB coactivator [112]. Thus, both Tat and NF-κB—the important regulators of HIV transcription—depend on P-TEFb.

The regulation of P-TEFb itself is also important for HIV transcription. *CycT1* expression is post-transcriptionally regulated and depends on specific miRNAs that block the translation of *CYCT1* mRNA by binding to its 3′ UTR [112]. Cell activation relieves this inhibition, causing a rapid rise in P-TEFb levels [113]. Nevertheless, most P-TEFb is sequestered and inactivated in the 7SK small nuclear ribonucleoprotein (7SK snRNP) [114]. MePCE and La-related protein 7 (LARP7) bind and protect the 5′ and 3′ ends of 7SK small nuclear RNA (snRNA) from degradation. Furthermore, the 5′ RNA stem-loop of the 7SK snRNP resembles TAR [114] and is bound by cellular hexamethylene bis-acetamide–inducible proteins (HEXIM 1 and 2), which undergo a conformational change upon RNA binding. RNA-associated HEXIMs then bind and inhibit P-TEFb in the 7SK snRNP. The rapid release of P-TEFb from the 7SK snRNP also increases the synthesis of HEXIM1, which reassembles the 7SK snRNP and sequesters free P-TEFb [87,115].

The ratio of free and bound P-TEFb (the P-TEFb equilibrium) then regulates the state of cell growth, proliferation, and differentiation [87]. Although cell activation increases the levels of P-TEFb sufficiently to support HIV transcription [116], transcription is further boosted when P-TEFb is rapidly released from the 7SK snRNP. This release follows cell stress from apoptosis, radiation, UV light, and compounds that inhibit transcription (e.g., actinomycin D, DRB, or flavopiridol); chemicals that produce changes in chromatin (such as histone deacetylases, or HDACs) and bromodomain and extraterminal (BET) inhibitors; and those affecting DNA methylation (5-azacytidine) [116]. Notably, these latency reversing agents (LRAs) only work when P-TEFb is abundant and present in the 7SK snRNP [116].

HDAC1 causes the chromatin remodeling responsible for inhibiting HIV-1 gene expression [117]. CBF-1, CTIP2, recombinant YY1 transcription factor (YY1), and late SV40 factor (LSF) are also thought to be implicated in the recruitment of HDACs to the pro-viral promoter and in the establishment of latency [87,107] (Figure 3). Furthermore, CTIP2 has been reported to recruit HDAC1 and HDAC2 to the 5′LTR of HIV-1 pro-viral DNA in microglia [52]. CTIP2 also interacts with SUV39H1, a methyl transferase responsible for H3K9me3, which subsequently promotes the recruitment of HP1 protein to the 5′LTR, leading to localized heterochromatization and latency (Figure 3).

A key study demonstrated that CTIP2 expression in the postmortem brain tissue from HIV+ controls, latent HIV+, and HIVE cases found higher levels of CTIP2 in the latent HIV+ cases compared to HIV+ controls and HIVE cases. Using double labeling techniques, the presence of CTIP2 was found only in the microglial cells of latent HIV+ cases. These data provide further support for the role of CTIP2 in regulating latency in microglial cells [75,117]. Furthermore, a role for lysine-specific demethylase (LSD1) in synergistically regulating HIV gene expression along with CTIP2 in microglia has also been described [118].

LSD1 aids in the recruitment of CTIP2 and hSET1/WDR5 (members of hCOMPASS complex) at the Sp-1 binding sites of the HIV proximal promoter, thereby increasing H3K4 trimethylation (H3K4me3) [118] (Figure 3), leading to increased repression of the viral gene. Furthermore, a decrease in H3K4me3 upon reactivation release of LSD1 and hSET1/WDR5 from the viral LTR has been reported, further confirming the role of these factors in modulating latency in microglia [118]. The diverse range of functions of LSD1 in microglia indicate its involvement as an anchor protein which assists in the recruitment of other factors at LTR [118].

In support of this, another study described the establishment of simian immunodeficiency virus latency in the macaque brain by an interplay of interferon-beta and dominant-negative CCAAT/enhancer-binding protein-beta (C/EBP-beta) isoforms, resulting in histone acetylation restriction and suppression of LTR activation. Thus, given that microglia are a long-lived latent reservoir of HIV-1 in infected individuals, targeting factors such as CTIP2, which are implicated in the regulation of latency, could be a reasonable therapeutic approach for these patients [119].

The presence of multiple T-cell factor 4 (TCF-4) binding sites have previously been shown in the 5′LTR of pro-viral DNA. Moreover, generation of protein complexes, including TCF-4, beta-catenin, and SMAR1 (a nuclear matrix binding protein) at position −143 on the 5′LTR, has been reported to inhibit pro-viral gene expression in astrocytes [120]. Of note, monocyte and macrophage cells have undamaged beta-catenin signaling [87]. Thus, the function of these proteins in inducing latency in these lineages has also been hypothesized.

#### 5.2.4. MicroRNA and HIV Latency in HAND

The main role of miRNAs is to regulate the effects of virus propagation and replication. Thus, they are important agents in gene silencing and in post-transcriptional protein regulation. HIV-1 is produced by viral genes involved in pathogenesis that interact with the host cell mRNAs [6,121]. In vitro, HIV-1 blocks the expression of miRNA cluster miR-17/92 to efficiently replicate in peripheral blood mononuclear cells (PBMCs), monocytes and macrophages [122]. Of note, monocytes are less susceptible to HIV-1 infection than macrophages and, thus, the presence of higher levels of anti-HIV miRNA (miRNA-382, miRNA-223, miRNA-150, and miRNA-28). In addition, elimination of these anti-HIV miRNAs in monocytes increases in HIV replication in these cells, while the use of miRNA mimics in MDMs decreases HIV replication [123].

Another original work also studied the differentially expression in the brains of the miRNAs of persons living with HAND. These four miRNAs (miR-500a-5p, miR-34c-3p, miR-93-3p, and miR-381-3p) have been analyzed to regulate expression of the peroxisome biogenesis factors 2, 7, 11B, 13, and 19 (PEX2, PEX7, PEX11B, PEX13, and PEX19). Subsequent analyses indicated that elevated expression of these miRNAs is very common in HIV infection. These data demonstrate that the raised levels of mRNAs that downregulate peroxisomes can act as a process to change antiviral signaling that emanates from these organelles [124,125,126].

There are numerous peroxisome-based diseases in humans, including different serious diseases such as Zellwegger syndrome spectrum disorders and rhizomelic chondrodysplasia punctata, and leukodystrophy (inflammatory degeneration of white matter), similar to that observed in advanced HAND, termed HIV-associated dementia [124]. These data highlight the fact that even partially diminished function of peroxisomes can produce a severe neurological disease.

Activation of mitochondrial antiviral signaling (MAVS)-dependent signaling from peroxisomes by different RNA viruses produce activation of type III interferon. MAVS signaling from both peroxisomes and mitochondria is required for maximal anti-viral activity. Another pivotal process that involves PEX19 (crucial for degradation) is produced by flaviviruses in the West Nile virus and Dengue virus, a critical peroxisome biogenesis factor. Flavivirus-infected cells contain significantly lower numbers of peroxisomes that drastically reduce type III interferon. Human cytomegalovirus HCMV involves protein vMIA and interacts with peroxisomal MAVS and produces fragmentation. The disruption of peroxisomal morphology is not essential for this viral protein, and the protein requires interaction with PEX19 to block antiviral signaling [124].

Another work found three miRNAs to be elevated in both brain tissue from rhesus macaques with and without simian immunodeficiency virus encephalitis (SIVE) and HIVE (miR-142-5p, miR-142-3p, and miR-21) [125,126]. Further analysis revealed that miR-21 also stimulated N-methyl-D-aspartic acid receptors and targeted the mRNA of myocyte enhancer factor-2C (MEF2C). Similarly, Noorbakhsh et al. (2010) identified the differential expression of multiple miRNAs in frontal lobe white matter by comparing HIV-negative and HIVE cases matched by age and sex [127], using the standard two-fold cut-off as the threshold for further analysis in their expression profiling study. Thus, given all these results, strategic miRNA manipulation could provide a potent therapeutic tool against HAND.

Latent HIV-1 reservoirs represent the largest roadblock to the complete eradication of HIV-1 from infected individuals. However, according to the ‘Kick and Kill’ strategy, viruses can be activated in these reservoirs using a range of latency-reversing agents, including histone deacetylase inhibitors (HDACi), histone methyltransferase inhibitors (HMTi), DNA methyltransferase inhibitors (DNMTi), protein kinase C (PKC) agonists, and several other small molecules. The impact of these LRAs has been well studied in CD4^+^ T cells and, to a lesser extent, in monocyte and macrophage lineage cells. Upon reactivation, latent HIV-1 undergoes robust replication, resulting in the production of an enormous amount of virus that can induce the lysis of target cells or infected cells and can be recognized by the cellular immune clearance machinery. In addition, fresh infection can be stopped by cART. Nonetheless, the impact of LRAs in reactivating latent virus in monocyte and macrophage lineage cells is not yet well studied and requires further investigation.

## 6. Targeting the HIV-1 Reservoirs as Innovative Approaches against HAND

Several methods have been studied as potential strategies to cure HIV, including targeting the CD4 T cell reservoir compartment, which can be redirected to other cellular HIV viral sanctuaries such as a microglial cells (Figure 4).

### 6.1. Early and Intensified Antiretroviral Therapy

With early initiation soon after infection, cART is considered a therapy able to achieve HIV remission. The VISCONTI Study Group cohort of adults with controlled HIV were able to spontaneously control HIV replication following the cessation of cART [128]. This event was also transiently shown in the infant referred to as ‘the Mississippi baby’ who received cART 30 h after birth. Current work studying very early cART intervention in several cohorts has also provided data supporting this curative strategy [129]. Nevertheless, viral rebound did take place in the Mississippi baby, indicating that early cART introduction alone may be insufficient as a therapy. Ongoing studies by Sacha et al. are evaluating the myeloid reservoir soon after infection and assessing the efficacy of early mega cART treatments in patients caught in the early acute stages of infection [130].

ART is unable to sufficiently suppress circulating virus-infected monocytes or macrophages, and the CNS compartment still requires further investigation. Several in vitro studies have shown that the current cART drugs have differing efficacies against HIV infection in non-T cell HIV reservoirs. Indeed, both maraviroc and raltegravir have been tested under cART intensification. Ndhlovu et al. showed that intensification with maraviroc for 24 weeks in HIV-infected, cART-treated participants reduced monocyte HIV DNA levels [131]. Another study showed no appreciable proviral DNA reservoir reduction in 10 PBMC and gut biopsy samples from HIV-infected patients either on emtricitabine/tenofovir or lopinavir/ritonavir, with further intensified antiretroviral therapy with maraviroc and raltegravir, except in a few patient cases with a modest reduction in proviral DNA in the gut [132]. This and other studies may be useful to develop a systematic clinical intervention sequence for ART therapies designed for non-T cell-HIV reservoirs.

### 6.2. Immune-Based Strategies

The use of cytokines and other agents that revert negative regulation of immune activation are also considered good immune-based therapies because of their capacity to both antagonize virus silencing and restore immune functions. It has not yet been established whether reactivated cells are killed by cytopathic effects or recognized and eliminated by the immune system [133]. Cytokines including IL-2, IL-7, and IL-15 with putative roles in HIV-1 control are currently being tested in humans. However, further studies will be required to elucidate the effective potential of these cytokines as candidates to deplete the HIV reservoir. In addition, preventative HIV vaccine trials have generally been ineffective, with only one clinical study so far having shown a protective effect, although there was no noticeable effect on the degree of viremia or CD4^+^ T cell numbers in patients who were eventually infected [134].

Furthermore, there is evidence that reversal of HIV latency may not necessarily lead to clearance of reactivated cells, with several therapeutic vaccine strategies having been tested in this context [135]. Therapeutic vaccination could re-stimulate CTL (CD4^+^ T cells) responses, mimicking the situation of patients in which viral replication is controlled without treatment and without progression to AIDS [136]. This method may be useful in reactivating the virus because the HIV-1 env and pol antigens were reported to activate most of CD8^+^ T cells harboring proviral DNA, inducing HIV-1 replication [136]. In the NCT00107549 clinical trial, patients were immunized with a poxvirus vaccine engineered to express HIV-1 antigens. A significant, albeit transient, decrease in replication-competent HIV-1 in the resting T-cell reservoir was observed in these patients [137].

Some vaccination treatments and pathogen infections have been found to transiently increase viral RNA in the plasma of HIV-1-infected patients during ART. In fact, the use of TLR agonists have recently been suggested to both reactivate HIV-1 from latently infected cells and to boost HIV-specific cytotoxic CD8^+^ T cell immunity [138]. On the contrary, the use of TLR antagonists such as chloroquine is thought to lower immune activation. Treatments with hydroxychloroquine or chloroquine have been evaluated in the clinical setting in ART-treated [139] and ART-naïve patients [139] and was reported to reduce immune activation, albeit with little or no effect on CD4^+^ T cell recovery. However, in a randomized trial in naïve, non-progressing patients, hydroxychloroquine treatment resulted in the worsening of CD4^+^ T cell loss [140].

Strategies aimed at the recovery of immune system functionality are pivotal in current studies. An important immunological target is programmed cell death protein 1 (PD-1), a receptor known for its role in immune exhaustion. PD-1 receptor is upregulated in HIV-specific and non-specific CD4^+^ and CD8^+^ T cells, limiting their functions [141]. Moreover, the compartment of memory T cells expressing elevated levels of PD-1 contain more proviral HIV-1 DNA than PD-1 low cells [141]. In patients on cART, a consistent association between the frequency of PD-1-expressing cells and the size of the reservoir has been shown [142]. Indeed, PD-1 is expressed at lower levels in “elite controllers” and “long-term non-progressors” (LTNPs) compared to typical progressors [143]. These recent data indicate that PD-1 high cells form a crucial reservoir for the virus and that PD-1 inhibition may also support beneficial effects by reducing the latent reservoir.

Some work has also attempted to test whether blocking this negative regulator of immune activation may both activate HIV-1 transcription and reverse immune exhaustion by relieving a functional block on virus-specific CD8 memory T cells [144]. Indeed, in vivo studies testing anti-PD1 antibodies on non-human primate models of HIV infection have shown positive effects on immune system restoration. These findings improved immune responses in the blood and gut and were associated with significant reductions in plasma viral load, greater antibody responses to both SIV and non-SIV antigens, and prolonged survival of SIV-infected macaques [145]. Furthermore, PD-1 blockade reduced persistent immune activation and showed evidence of the restoration of mucosal barrier integrity and bacterial translocation [145]. A study by an AIDS clinical trial group (ACTG 5301) evaluating the safety and efficacy of PD-1 blockade to reduce the latent HIV reservoir in patients in cART is currently being developed.

The antifibrotic effects of angiotensin-converting enzyme inhibitors and angiotensin receptor blockers through suppression of transforming growth factor-β in different clinical settings has more recently been studied in the context of HIV infection, with clinical trials is currently in phases I and II (NCT01535235; ACTG 5317). Their capacity to improve HIV-specific immune responses and to reduce the viral reservoir in lymphoid tissues is also being assessed. Two other promising anti-inflammatory molecules are the peroxisome proliferator-activated receptor agonists pioglitazone and leflunomide, which are both well tolerated and effective in the treatment of chronic inflammation and reduce the metabolic syndromes associated with prolonged ART [146].

### 6.3. Stem Cell Transplantation

An important case report by Allers et al. demonstrated the transplant of homozygous CCR5 delta32 stem cells into an HIV-1 infected patient that suffered acute myeloid leukemia, resulting in a solution for the disease whereby the “Berlin patient” was able to stop antiretroviral treatment with viremic rebound [147] (Figure 4). No virus was identified in multiple samples of non-CD4 T cell populations from this patient when using different assessment techniques [148]. Nevertheless, it remains undetermined whether this stem cell transplantation strategy was able to overcome the HIV reservoir in macrophages and microglia. Of note, the prior administration of gemtuzumab ozogamicin (an anti-CD33 monoclonal antibody) in this patient may have targeted myeloid cells for depletion (Figure 4).

A second patient (the “London patient”) with an analogous medical scenario was also apparently cured from HIV-1 infection [149]. However, clinicians still do not know whether a lack of myeloid-targeted depletion contributed to incomplete reservoir removal and, thus, viral reemergence in other subsequent human and non-human stem cell transplantation cases (Figure 4). Stem cell transplantation remains an understudied and multifaceted strategy for treating HIV infection but understanding and clearing the latent viral reservoir remains a key area for study in this field.

### 6.4. Permanent HIV Suppression

Activation of p-TEFb has recently been established as the mechanism by which the LRAs, vorinostat and panobinostat, work to activate latent HIV in CD4^+^ T cells [150]. Nevertheless, because p-TEFb is required for RNAPII activation and therefore viral production, p-TEFb may also be a target for permanent HIV-1 suppression. Heat shock protein 90 (Hsp90) inhibitors can block the NF-kB pathway and thus inhibit HIV-1 transcription [151]. In addition, PIM-1 inhibitors can block HIV-1 reactivation, making it a possible permanent suppression therapy [129,151]. However, whether these novel approaches can be extended to non-CD4 T cell reservoirs remain untested and should be further evaluated as an alternative strategy.

### 6.5. Gene Therapy and the “Kick and Kill” and “Shock and Kill” Therapeutic Strategies

The CRISPR-Cas9 system is a novel tool that uses guide RNA instead of custom proteins to home in on target DNA [129,152] (Figure 4). It is based on nuclease-mediated gene editing tools such as zinc finger nuclease (ZFN1) transcription activator-type effector nucleases (TALENs) and CRISPR/Case technologies [153]. One major challenge for both ZFN and TALENs is the expense and labor involved. Zhang et al. reported using a dCas9-synergistic activation mediator (dCas9-SAM) system to reactivate HIV-1 in both CD4^+^ T cell and microglial cell lines [154]. The possible advantage of this system is that it may minimally affect localized HIV-negative cells. However, the most promising results were presented by Hu et al. who developed a Cas9/guide RNA system to eradicate the HIV-1 genome and immunize target cells against HIV-1 reactivation in latently infected microglial, promonocyte, and T cell lines [155] (Figure 4). Reactivation of HIV expression in CD4^+^ T cells and in microglial cell lines was also demonstrated in the study by Cary and Peterlin [156] who also used CRISPR/dCas9 to synergistically reactivate HIV when related to HDAC inhibitors and PKC activators [157]. Therefore, Cas9 gene therapy systems may need to be tailored to each compartmentalized viral reservoir.

The “Shock and Kill” strategy, “Block and Lock” strategy, and microglial cell gene therapy, which could all provide a functional cure, were proposed by Wallet et al. [8] (Figure 4). Of these three strategies, the former two are pharmacological methods, while the latter is based on CRISPR/cas9- technology as a gene editing tool. The “Shock and Kill” cannot remove reservoirs such as microglial cells because their reactivation will drive neuroinflammation and exacerbate HAND. Potential epigenetic regulators that control microglial cells are therefore required. To date, the most examined epigenetic regulators are long non-coding RNAs [158] and miRNAs [159]. Nevertheless, due to the stochasticity of HIV-1 transcription, the Shock and Kill method does not favor reactivation of all latently infected microglial cells [8].

In contrast, “Block and Lock” therapy is a very convenient approach because it is associated with a drastically reduced risk of brain inflammation compared to “Shock and Kill.” This strategy could be used following the “Shock and Kill” strategy because it targets transcription and/or RNA export to counteract the effects of proinflammatory cytokines and prevent synthesis of viral proteins [160]. Thus, the combination of these two approaches could drive a reduction in the microglial cell reservoir, although it may cause deep latency in reservoirs that are not reactivated by LRAs.

In turn, CRISPR/Cas9 technology is a novel and elegant technique that operates on specific sequences and can involve cellular or viral factors. This strategy can be used either to reactivate the virus or to excise and eliminate the provirus from its host genome [8]. However, several questions remain unclear, such as whether it can be administered in vivo in the brain and its potential long-term toxicity. Therefore, more work will be required before it can be clinically applied. In vitro models have demonstrated that a novel and non-invasive approach to administering CRISPR/cas9-guided RNA across the BBB inhibits latent microglial cells infection [161]. Nonetheless, we must gain a more thorough understanding of the specific molecular mechanisms of HIV-1 latency in microglia to be able to design new molecules and new strategies to prevent HAND. Furthermore, new therapies to circumvent the limitations related to anatomical sanctuaries with barriers (such as the BBB) that limit the entry of drugs are still needed.

### 6.6. Progress and Limitations of Treating HAND by Targeting Microglia

Recent progress in the treatment of HAND by targeting microglia has been described. This work reveals that viral envelope glycoproteins can fuse only with cells expressing both CD4 and an HIV co-receptor, with CCR5 being the most important co-receptor in microglia and CNS macrophages [160]. CCR5 receptors are upregulated on activated CD4^+^ and CD8^+^ T cells, and enhance antigen-presenting cell interactions, T cell trafficking into tissues, and cytokine production [162,163]. In response to infection or inflammation, monocytes, microglia, astrocytes, and neurons express CCR5 ligands, which increase the migration of CCR5+T cells into the CNS [161]. Locally, these effector T cells secrete CCR5 ligands, which amplify the immune response [162,163]. CCR5-tropic viruses have been found to be highly fusogenic, and lower expression levels of CCR5 and CD4 are required to infect cells, thus causing increased apoptosis [164]. There is confirmation that the plurality of HIV strains in the brain are all CCR5 tropic [164]. Due to the intrinsic role of CCR5 in CNS disease, CCR5 antagonists such as maraviroc may be able to reduce the migration and activation of effector CD8^+^ T cells in the CNS and hence reduce the neurocognitive sequelae of HIV [158].

Maraviroc has satisfactory CSF penetration and low rates of resistance, inhibits CNS viral replication, and possesses anti-inflammatory properties in the CNS [158]. This drug blocks HIV entry into cells, therefore restricting virion replication and indirectly inhibiting immediate post-integration viral protein production, which probably drives the CNS inflammatory milieu [163]. A recent prospective, open-label pilot randomized controlled clinical trial in patients with viral suppression and stable ART for 12 months found maraviroc-intensified ART improved global neurocognitive performance at both 6 and 12 months without significant unwanted secondary effects [164]. In the future, large randomized controlled studies will be necessary to ratify these data. Nevertheless, this suggests that maraviroc intensification of ART could be an alternative for the clinical management of HAND occurring in the context of viral suppression.

Furthermore, an additional emerging area of interest is microRNAs, the non-coding RNA molecules which regulate gene expression, with miR125b and 146a playing important roles in microglial infection and cell death [165]. In a small-scale study of 10 patients (9 with HAND), these microRNAs correlated with HIVE. Nevertheless, the interaction with ART and viral suppression in this context has yet to be explained in detail [166].

Other recent studies have aimed to evaluate the effects of cART-mediated oxidative stress on the induction of inflammatory responses in microglia. Tripathi et al. chose three ART drugs—tenofovir disoproxil fumarate, emtricitabine, and dolutegravir—and replicated previous findings that exposure of microglia to these cART cocktails caused the production of ROS, subsequently leading to lysosomal dysfunction and dysregulated autophagy, ultimately resulting in microglial activation. Interestingly, the potent antioxidant, N-acetylcysteine, could reverse the deleterious effects of cART [167].

These in vitro data were corroborated in vivo when cART-treated HIV transgenic rats demonstrated increased prefrontal cortex microglial activation, exaggerated lysosome impairment, and dysregulated autophagy compared to HIV transgenic rats not exposed to cART. Furthermore, the treatment of HIV transgenic rats with N-acetylcysteine also reduced the damaging effects of cART. Therefore, these data indicate that oxidative stress-mediated lysosomal dysfunction has an important function in the pathogenesis of HAND in drug-treated HIV-infected patients and that antioxidant-mediated mitigation of oxidative stress could be considered an adjunctive therapeutic strategy for alleviating or mitigating some of the neurocognitive problems of HAND [167].

## 7. Combined Antiretroviral Therapy Treatment for HAND

Combined antiretroviral therapy has significantly improved the quality of life of HIV-1 patients, decreasing the morbidity and mortality related to HIV/AIDS and transforming this disease into a chronic illness requiring lifelong treatment. However, cART still cannot remove the latent reservoir to control persistent replication in macrophages and microglia. Indeed, Borrajo et al. (2017) and Eisele et al. (2012) demonstrated that virus revival from latent reservoirs occurs once patients stop treatment [2,21]. A recent study showed that the rise in new cases of HAND is due to HIV-1 infection itself and not infection by other opportunistic pathogens [168]. At present, there is no definitive cure for HAND, with the only option remaining strong adherence to antiretroviral therapy to maintain a low viral load in the blood [169]. Combined antiretroviral therapy has been shown to help to improve the brain function of people living with HIV-1 [170]. Nonetheless, there is currently controversy about the real protective effects of antiretroviral drugs on the CNS and the potential adverse effects of using this therapy.

Some studies have shown a decrease in cognitive damage due to the use of cART, with people with a controlled viral load who discontinued cART exhibiting an improvement in cognitive functions and decreased neuronal impairment [171]. Several studies have reported that the complications of therapy are largely associated with the secondary effects of some antiretrovirals such as efavirenz, stavudine, zidovudine, and abacavir, which are related with neurological disorders, including HAND [85]. This latter study reported that efavirenz is related to insomnia, anxiety, and depression during the first week of use [85]. Secondary effects such as these can result in lower patient adherence to treatments.

Finding a cure or strategy to combat the latent reservoir of HIV-1 remains a priority. However, this is a challenge because of the characteristics of the virus, including its high mutation rate and ability to adapt to allow HIV-1 to enter or act on different types of target cells [85]. In addition, drugs used to treat HIV-1 reach a range of concentrations in different cells in the CNS [85,157]. Moreover, the concentration of the different drugs included in cART depends on the composition of the treatment employed [85], meaning that each drug may achieve diverse concentrations in the same cell, making it difficult to ensure the optimal functionality of these drugs.

Previous work has focused on influencing viral replication, inflammation, oxidative stress, neurotoxicity, neuroprotection, and apoptosis in patients with HIV [159]. Another current study examined the effects of two immunosuppressive and anti-inflammatory drugs—teriflunomide (Teri) and monomethylfumarate (MMF)—on monocyte and microglial activation and neurotoxicity. This group demonstrated that Teri and MMF reduced the secretion of chemotactic and proinflammatory cytokines (CCL2, CCL5, and CXCL10) in a coculture system of microglia with HIV-transduced monocytoid cells, as shown by the decreased neurotoxicity of this supernatant in human fetal neurons [160] and reviewed in [6].

Many other medications have been studied as adjuvants for therapies to combat neurocognitive damage associated with HIV-1 infection. For example, recent research has demonstrated that an insulin treatment inhibited HIV-1 supernatant p24 levels and decreased CXCL10 and IL-6 transcript levels in HIV-infected primary human microglial cells [162]. Of note, this might be associated with metabolic and trophic effects and could help protect neurons and reduce inflammatory cytokine expression [172]. Thus, this new treatment represents a new therapeutic opportunity for patients with HAND.

## 8. Conclusions

In recent years, there has been a significant reduction in deaths associated with HIV-1 thanks to early cART treatments. However, complete cure is not possible without targeting latent viral reservoirs. Microglial cells play a vital role as latent reservoirs because they are resistant to the cytopathic effects of the virus and have a long lifespan. Therefore, they can disseminate HIV-1 for longer periods of time. Moreover, microglial cells represent anatomical viral sanctuaries with poor cART penetration. Containing and/or eliminating these viral reservoirs remains a daunting problem for clinical science. Several studies have focused on reactivating latent reservoirs in the presence of the optimal cART, improving cytotoxic response, and inducing apoptosis of these infected cells. Whereas it is particularly important to target microglial cells, this is an incredibly challenging prospect. However, combining these emerging approaches with well-established cART treatments may be able provide the so-far elusive cure for HAND by targeting infected microglial cells.

## Figures and Tables

**Figure 1 biomedicines-09-00925-f001:**
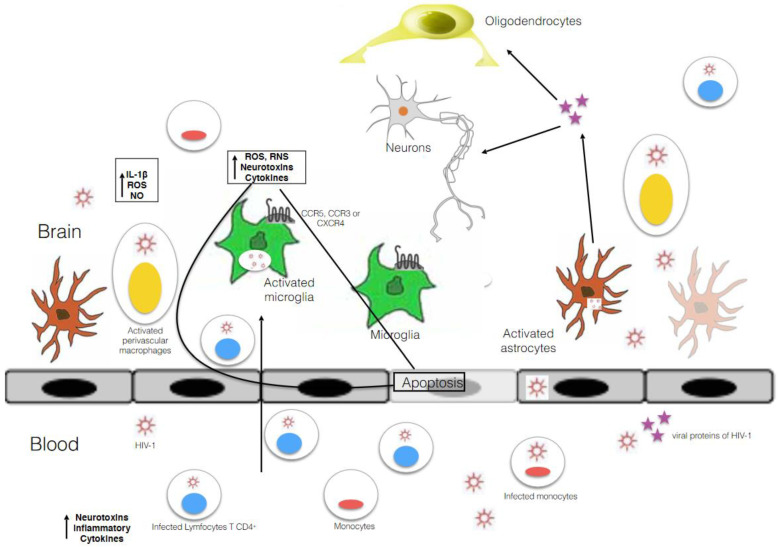
Schematic representation of the entry of HIV and its viral proteins into the brain. HIV enters the perivascular space (the main site for viral replication) by migration across the BBB via infected macrophages or blood lymphocytes or as free virions (viral particles) where it infects and activates macrophages, astrocytes, and microglia. The activation of these cells plays a key role in the release of proinflammatory cytokines and can amplify the alteration and permeability of the BBB. HIV-1 envelopes glycoprotein; attaches the virion to macrophages, astrocytes, and microglia; and induces the fusion of viral and cell membranes to initiate infection. It interacts with the chemokine receptor CCR5 or CXCR4 to allow viral entry by triggering large structural rearrangements and unleashing the fusogenic potential of gp41 to induce membrane fusion.

**Figure 2 biomedicines-09-00925-f002:**
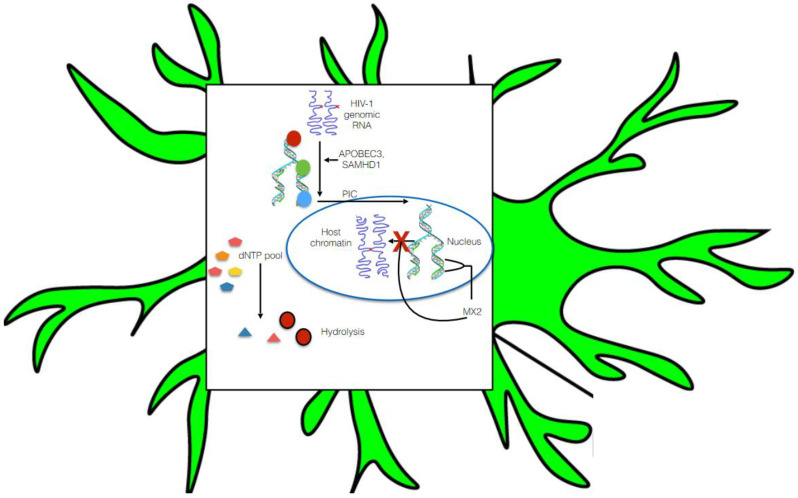
Model of pre-integration latency in HIV-1-infected microglia. Pre-integration latency is controlled by interaction of host restriction factors, including APOBEC3, SAMHD1, and MX2 in the monocyte- or macrophage-lineage cells. APOBEC3 plays a pivotal role in causing G to A hypermutation of the HIV-1 genome. In particular, SAMHD1 reduces the pool of dNTPs in macrophages by hydrolyzing them into their precursors (nucleosides and triphosphates), resulting in ineffective viral reverse transcription. Furthermore, MX2 restricts HIV replication in macrophages and microglia. Moreover, MX2 blocks HIV infection at the post-entry level by hindering the nuclear accumulation and integration of pro-viral DNA into host chromatin.

**Figure 3 biomedicines-09-00925-f003:**
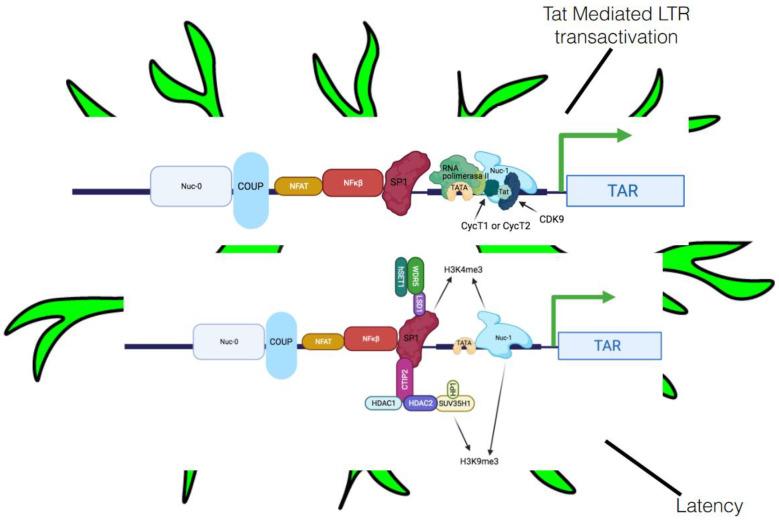
A model of post-integration latency in HIV-1-infected microglia. Post-integration latency is manifested through several mechanisms, including chromatin remodeling, epigenetic mechanisms, and host-encoded miRNAs. Histone deacetylase 1 (HDAC1) causes chromatin remodeling, which is responsible for the inhibition of HIV-1 gene expression. C-promoter binding factor-1 (CBF-1) and COUP-TF-interacting protein 2 (CTIP2) have also been implicated in the recruitment of HDACs to the pr-oviral promoter, thereby establishing latency. Moreover, Tat can mediate reactivation of this process. Integrated HIV-1 pro-viral DNA is also delimited at both ends by the HIV-1 LTR. The 5′ LTR has unique blend of a robust TATA box, a potent initiator sequence, and binding sites for several TFs, including the glucocorticoid receptor COUP, upstream stimulatory factor (USF), activator protein 1 (AP1), TCF-1α, C-Myc, specificity protein 1 (SP1), CTF/NFE, nuclear factor of activated T cells (NFAT), nuclear factor kappa-light-chain-enhancer of activated B cells (NF-κβ), and Tat.

**Figure 4 biomedicines-09-00925-f004:**
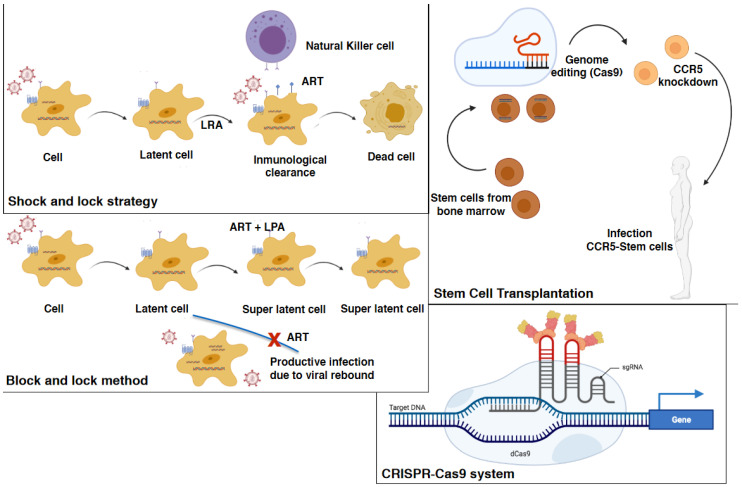
Innovative strategies against HAND. In the “Shock and Kill” approach, integrated pro-viral DNA is kicked into transcriptional activity by an LRA. The virally active cells will subsequently be eliminated with the continuation of antiretroviral therapy during the ‘kill’ phase. Several molecules, including epigenetic modifying agents such as HDACi, HMTi, and DNMTi, have been shown to reactivate the virus in vitro, ex vivo, and in vivo. The ‘Block and Lock’ strategy toward a functional cure aims to permanently silence the latent reservoir by utilizing latency-promoting agents (LPAs) to ‘block’ virus transcription and ‘lock’ virus promoters in a latent state via repressive epigenetic modifications. Stem cell transplantation involves a homozygous CCR5 delta32 stem-cell transplant into an HIV-1 infected patient with acute myeloid leukemia with the removal of antiretroviral treatment without subsequent viremic rebound. In the CRISPRCas9 system, dCas9-SAM minimally affects localized HIV-negative cells in latently infected microglial cells, promonocytes, and T cell lines. A Cas9/guide RNA system has been developed to eradicate the HIV-1 genome and immunize target cells against HIV-1 reactivation.

**Table 1 biomedicines-09-00925-t001:** Studies about mechanisms in microglial cells.

Analyzed Mechanisms	Study	Reference
Function of HIV-Tat in microglia	E.T. Chivero, M.-L. Guo, P. Periyasamy, K. Liao, S.E. Callen, S. Buch, HIV-1 Tat primes and activates microglial NLRP3 inflammasome-mediated neuroinflammation (2017)	[16]
Detection of infected microglia in patients whose viral level is suppressed but died from an HIV-1 unrelated outcome	A. Ko, G. Kang, J.B. Hattler, H.I. Galadima, J. Zhang, Q. Li, W.K. Kim, Macrophages but not astrocytes harbor HIV DNA in the brains of HIV-1-infected aviremic individuals on suppressive antiretroviral therapy (2019)	[23]
SAMHD1 restriction take place in in vitro differentiated macrophages and in freshly isolated macrophages from the lungs, abdomen, and brain	J.J. Cenker, R.D. Stultz, D. McDonald, Brain Microglial Cells Are Highly Susceptible to HIV-1 Infection and Spread (2017)	[24]
Over stimulation of microglia contributes to HAND	R. Geffin, R. Martinez, R. Perez, B. Issac, M. McCarthy, Apolipoprotein E-dependent differences in innate immune responses of maturing human neuroepithelial progenitor cells exposed to HIV-1 (2013)	[36]
Increased release of proinflammatory chemokines by microglia	S.L. Wesselingh, K. Takahashi, J.D.Glass, J.C. McArthur, J.W. Griffin, D.E. Griffin, Cellular localization of tumor necrosis factor mRNA in neurological tissue from HIV-infected patients by combined reverse transcriptase/polymerase chain reaction in situ hybridization and immunohisto-chemistry (1997) D. Alvarez-Carbonell, Y. Garcia-Mesa, S. Milne, B. Das, C. Dobrowolski, R. Rojas, J. Karn J, Toll-like receptor 3 activation selectively reverses HIV latency in microglial cells (2017)	[56,58]
HIV-1 replication in microglial cells and their activation are dependent on autophagy activation	S.S. Choi, H.J. Lee, I. Lim, J. Satoh, S.U. Kim, Human astrocytes: secretome profiles of cytokines and chemokines (2014)	[57]
HIV-1 replication in microglia using primary cultures obtained from fetal brains or post-mortem assessment of AIDS patients	A.V. Albright, J.T. Shieh, M.J. O’Connor, F. Gonzalez-Scarano, Characterization of cultured microglia that can be infected by HIV-1 (2000) S. Peudenier, C. Hery, L. Montagnier, M. Tardieu, Human microglial cells: characterization in cerebral tissue and in primary culture, and study of their susceptibility to HIV-1 infection (1991)	[58,59]
Immortalized human microglial cell line is useful to produce stable cell lines latently infected with HIV-1 proviruses	Y. Garcia-Mesa, T.R. Jay, M.A. Checkley, B. Luttge, C. Dobrowolski, S. Valadkhan, G.E. Landreth, J. Karn, D. Alvarez-Carbonell, Immortalization of primary microglia: A new platform to study HIV regulation in the central nervous system (2017) D. Alvarez-Carbonell, F. Ye, N. Ramanath, C. Dobrowolski, J. Karn, The glucocorticoid receptor is a critical regulator of HIV latency in human microglial cells (2019)	[59,60]
Absence of restriction by SAMDH1 is due to its phosphorylation by the cyclin kinase 1 (CDK1) which is induced in microglial cells	P. Mlcochova, K.A. Sutherland, S.A. Watters, C. Bertoli, R.A. de Bruin, J. Rehwinkel, S.J. Neil, G.M. Lenzi, B. Kim, A. Khwaja, M.C. Gage, C. Georgiou, A. Chittka, S. Yona, M. Noursadeghi, G.J. Towers, R.K. Gupta, A G1-like state allows HIV-1 to bypass SAMHD1 restriction in macrophages (2017)	[61]
Identification of HIV-1 DNA, RNA and protein in microglial cells of brain autopsies from patients that died from severe form of HAND	M.A. Cosenza, M.-L. Zhao, Q. Si, S.C. Lee, Human brain parenchymal microglia express CD14 and CD45 and are productively infected by HIV-1 in HIV-1 encephalitis (2002) M.J. Churchill, P.R. Gorry, D. Cowley, L. Lal, S. Sonza, D.F. Purcell, K.A. Thompson, D. Gabuzda, J.C. McArthur, C.A. Pardo, S.L. Wesselingh, Use of laser capture microdissection to detect integrated HIV-1 DNA in macrophages and astrocytes from autopsy brain tissues (2006)	[62,63]
Detection of HIV-1 DNAs in microglial cells and macrophages in brain autopsies from patients whose infection was controlled	K.A. Thompson, C.L. Cherry, J.E. Bell, C.A. McLean, Brain cell reservoirs of latent virus in presymptomatic HIV-infected individuals (2011)	[64]
Detection of the virus in the CSF in persons under effective cART, who had otherwise undetectable plasma HIV-1	A. Edén A, S. Nilsson, L. Hagberg, D. Fuchs, H. Zetterberg, B. Svennerholm, M. Gisslén, Asymptomatic cerebrospinal fluid HIV-1 viral blips and viral escape during antiretroviral therapy: a longitudinal study (2016)	[65]
Compartmentalization of HIV-1 in the CSF	D.F. Bavaro, A. Calamo A, L. Lepore, C. Fabrizio, A. Saracino, G. Angarano, L. Monno, Cerebrospinal fluid compartmentalization of HIV-1 and correlation with plasma viral load and blood–brain barrier damage (2019)	[66]
Macaques model with microglial cells infection is useful to conduct preclinical evaluation of therapeutic interventions aimed at eradicating HIV-1 from the CNS	C. Harbison, K. Zhuang, A. Gettie, J. Blanchard, H. Knight, P. Didier, C. Cheng-Mayer, S. Westmoreland, Giant cell encephalitis and microglial infection with mucosally transmitted simian-human immunodeficiency virus SHIVSF162P3N in rhesus macaques (2014)	[67]
Demonstration of the establishment of viral reservoir in animal models	J.B. Whitney, A.L. Hill, S. Sanisetty, P. Penaloza-MacMaster, J. Liu, M. Shetty, L. Parenteau, C. Cabral, J. Shields, S. Blackmore, J.Y. Smith, A.L. Brinkman, L.E. Peter, S.I. Mathew, K.M. Smith, E.N. Borducchi, D.I. Rosenbloom, M.G. Lewis, J. Hattersley, B. Li, J. Hesselgesser, R. Geleziunas, M.L. Robb, J.H. Kim, N.L. Michael, D.H. Barouch, Rapid seeding of the viral reservoir prior to SIV viraemia in rhesus monkeys (2014)	[68]
Infected microglia by HIV-1 in vivo in humanized mouse models	G.N. Llewellyn, D. Alvarez-Carbonell, M. Chateau, J. Karn, P.M. Cannon, HIV-1 infection of microglial cells in a reconstituted humanized mouse model and identification of compounds that selectively reverse HIV latency (2018) S. Mathews, A. Branch Woods, I. Katano, E. Makarov, M.B. Thomas, H.E. Gendelman, L.Y. Poluektova, M. Ito, S. Gorantla, Human Interleukin-34 facilitates microglia-like cell differentiation and persistent HIV-1 infection in humanized mice (2019) H. Su, Y. Cheng, S. Sravanam, S. Mathews, S. Gorantla, L.Y. Poluektova, P.K. Dash, H.E. Gendelman, Immune activations and viral tissue compartmentalization during progressive HIV-1 infection of humanized mice (2019)	[69,70,71]

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
