# Peer review of "Microglia: The Real Foe in HIV-1-Associated Neurocognitive Disorders?"

_biomedicines, 2021, doi:10.3390/biomedicines9080925_

Round 1
Reviewer 1 Report
The authors of the review article entitled ‘Microglia: real foe in HIV-1-associated neurocognitive disorders?, sum up recent progress in the field of combined anti-retroviral therapy (cART), open a new view on microglia cells roles. The literature cited is of important relevance, the authors are aiming to introduce their own style, interpretation as well as outlook to this interesting topic. Instead, the authors arrange a review that does hold an individual flavor. Further, Figures and tables are appealing, and original. The clinical question about the scientific opportunity to study the cART effects are too broad, but they are applicable to basically everything than characteristic for brain neurocognitive disorders. This is a well written review and the length of the paper is commensurate with the message. For the mentioned reasons, the manuscript does not needs of originality resulting in a review which guides the reader stringently through the story. This manuscript is acceptable for publication in present form.
Author Response
Thank you for your comments and opinion that allowed us to improve the quality of the manuscript. We are very grateful for your clever observations and we hope that we can continue to having a good feedback from you in future works. Thank you so much for everything.

Reviewer 2 Report
The review manuscript is extensive and covers the topic in a comprehensive and detailed way.
1. There is some redundancy, which does not necessarily improve the understanding if the topic. If revised, I recommend to try and reduce repetition of information, especially in the first part of the text.
2. The manuscript would also improve by having a separate short section about microglia biology.
3. The structure and writing style is not consistent throughout the text. It may substantially improve by having one author taking the lead and making sure that it is consistent in structure and style.
4. English scientific writing is deficient and there are grammatical mistakes. I recommend to have it revised by an expert of English mother tongue.
5. There are substantial similarities with other published texts in some sections! The authors should reword those parts.
Author Response
The review manuscript is extensive and covers the topic in a comprehensive and detailed way.
1. There is some redundancy, which does not necessarily improve the understanding if the topic. If revised, I recommend to try and reduce repetition of information, especially in the first part of the text.
Thank you for your very careful review of our paper, and for the comments, corrections and suggestions that ensued. We believe the paper has been significantly improved. In the final version of the manuscript, we have done the changes that you have suggested. We have improved understanding of the text, reducing repetition of information throughout all the text and eliminating those phrases or topics already discussed previously.
2. The manuscript would also improve by having a separate short section about microglia biology.
According with the clever Reviewer’s comment, we have added new text in a separate section about microglia's biology. The new text appears in red in the final version of manuscript (page 7, second paragraph up to 6th paragraph). Additional references also have been added in the paper (in red) (30, 31, 36).
Text added in the review (page 7, second paragraph up to 6th paragraph):
2. Microglia biology
Microglia are the resident immune cell population of the CNS and were discovered by Franz Nissl, Alois Alzheimer, Ramón y Cajal, and Pío del Río-Hortega during the 19th and 20th centuries [29]. Microglia, which can migrate, proliferate, and phagocytose, constitute some 10–15% of all the CNS cells and can be differentiated from neurons and other types of glial cells present in the brain by their morphological characteristics [26]. They establish numerous contacts with other CNS cells, including neurons, astrocytes, and oligodendrocytes.
Early studies indicated that, like neurons and glial cells, microglia had a neuroectodermal origin, however recent work has demonstrated that microglial cells are macrophages that belong to the immune system. Crucial studies in mice revealed that macrophages can be derived from different developmental pathways that differentially contribute to the respective tissue compartments in embryos and adults [30]. These cells come from myeloid precursors with ‘primitive’ and ‘definitive’ erythroid myeloid potential which are consumed before birth, as well as from hematopoietic stem cells that establish and maintain definitive adult lymphoid and myeloid hematopoiesis throughout adulthood [31]. Microglial cells originate from c-KitloCD41lo progenitors in the embryonic yolk sac early during embryogenesis [31] and have the potential for primitive erythropoiesis.
Microglia cells were first identified and ultrastructurally characterized in the corpus callosum in the mature brain [31]. Most of microglial cells in adult brain tissue are in a ‘resting’ (quiescent) state and typically have a branched morphology. In 1996, Kreutzberg et al. [32] used the in vivo morphology of microglia to classify them as (a) branched with small cells and many thin branches (quiescent) or (b) ameboid with truncated processes (an active status) to simplify proliferation, migration, and phagocytosis [33]. Nonetheless, quiescent microglia are not static, rather, their branches constantly move and work in mice has demonstrated that they dynamically expand and retract [34, 35].
The transformation of amoeboid microglia into ramified microglia is a process that coincides with the onset of axon myelination in the corpus callosum [31]. Electron microscopic studies have shown that amoeboid cells in 1 to 5-day-old animals possess a round nucleus with marginal chromatin clumps and abundant cytoplasm displaying lysosomal dense granules, vacuoles, and a well-developed Golgi apparatus. In contrast, most microglia in older animals are elongated and branched, have a flattened nucleus and scant cytoplasm containing only a few lysosomal granules [31]. These cells are immunoreactive with the microglia specific antibody OX42 and can also be labelled with lectin.
Accumulating evidence in recent years supports the idea that microglia are immune-regulatory cells that play important roles in the CNS in health and disease. They help maintain the homeostasis of the brain environment under normal conditions but also produce robust reactions in response to inflammatory stimuli, injury, hypoxia-ischemia, and other adverse conditions affecting the normal function of the brain. Activated microglia take on an amoeboid morphology, proliferate, and migrate to the site of injury or damage where they protect the CNS from viruses and pathogens [31]. However, over activation and/or chronic activation of microglia with the excess production of inflammatory mediators may result in neurotoxic consequences [36]. Indeed, microglial cells can contribute to neurocognitive degeneration, as seen in the different forms of HAND, leading to an increase in proinflammatory chemokines and cytokines as well as neurotoxins which can affect both astrocytes and neurons and can cause neuronal damage [37].
References (30,31,36):
[30] Ginhoux F., Guilliams M. (2016) Tissue-Resident Macrophage Ontogeny and Homeostasis. Immunity. 15;44(3):439-449. doi: 10.1016/j.immuni.2016.02.024.
[31] Prinz M., Jung S., Priller J. (2019) Microglia Biology: One Century of Evolving Concepts. Cell. 3;179(2):292-311. doi: 10.1016/j.cell.2019.08.053.
[36] Kaur C., Sivakumar V., Zou Z., Ling E.A. (2014) Microglia-derived proinflammatory cytokines tumor necrosis factor-alpha and interleukin-1beta induce Purkinje neuronal apoptosis via their receptors in hypoxic neonatal rat brain. Brain Struct Funct. 219(1):151-70. doi: 10.1007/s00429-012-0491-5.
3. The structure and writing style is not consistent throughout the text. It may substantially improve by having one author taking the lead and making sure that it is consistent in structure and style.
We have followed the reviewer’s recommendation and I have tried to improve the structure and style, doing it consistent throughout the text. Numerous changes can be seen in red throughout the text.
4. English scientific writing is deficient and there are grammatical mistakes. I recommend to have it revised by an expert of English mother tongue.
According with the important Reviewer’s comment, we have done several changes (in red throughout the text) in the structure, style (previously comment) and have corrected grammatical mistakes with the aid of an expert of English mother tongue.
5. There are substantial similarities with other published texts in some sections! The authors should reword those parts.
We have followed the reviewer’s advice and have rewritten the similar paragraphs with other text already published and have done important changes (in red) in the final version of the manuscript. We understand that is important error but this may be due to the different ways of working of each of authors involved in writing the review. We want to apologize for this trouble.
Text added (page 3; paragraph 6)
“Of note, microglia express receptors for various neurotransmitters as well as for innate immunity ligands, including pattern-recognition receptors such as toll-like receptors [6]. Furthermore, microglial cells present antigens and secrete cytokines [17]—low molecular weight proteins generally classified as pro- or anti-inflammatory types—involved in the physiological processes implemented to combat pathogens and repair tissues”
(page 7; paragraph 3)
“They establish numerous contacts with other CNS cells, including neurons, astrocytes, and oligodendrocytes. Early studies indicated that, like neurons and glial cells, microglia had a neuroectodermal origin, however recent work has demonstrated that microglial cells are macrophages that belong to the immune system. Crucial studies in mice revealed that macrophages can be derived from different developmental pathways that differentially contribute to the respective tissue compartments in embryos and adults [30]. These cells come from myeloid precursors with ‘primitive’ and ‘definitive’ erythroid myeloid potential which are consumed before birth, as well as from hematopoietic stem cells that establish and maintain definitive adult lymphoid and myeloid hematopoiesis throughout adulthood [31]. Microglial cells originate from c-KitloCD41lo progenitors in the embryonic yolk sac early during embryogenesis [31] and have the potential for primitive erythropoiesis.
Microglia cells were first identified and ultrastructurally characterized in the corpus callosum in the mature brain [31]. Most of microglial cells in adult brain tissue are in a ‘resting’ (quiescent) state and typically have a branched morphology. In 1996, Kreutzberg et al. [32] used the in vivo morphology of microglia to classify them as (a) branched with small cells and many thin branches (quiescent) or (b) ameboid with truncated processes (an active status) to simplify proliferation, migration, and phagocytosis [33]. Nonetheless, quiescent microglia are not static, rather, their branches constantly move and work in mice has demonstrated that they dynamically expand and retract [34, 35].
The transformation of amoeboid microglia into ramified microglia is a process that coincides with the onset of axon myelination in the corpus callosum [31]. Electron microscopic studies have shown that amoeboid cells in 1 to 5-day-old animals possess a round nucleus with marginal chromatin clumps and abundant cytoplasm displaying lysosomal dense granules, vacuoles, and a well-developed Golgi apparatus. In contrast, most microglia in older animals are elongated and branched, have a flattened nucleus and scant cytoplasm containing only a few lysosomal granules [31]. These cells are immunoreactive with the microglia specific antibody OX42 and can also be labelled with lectin.
Accumulating evidence in recent years supports the idea that microglia are immune-regulatory cells that play important roles in the CNS in health and disease. They help maintain the homeostasis of the brain environment under normal conditions but also produce robust reactions in response to inflammatory stimuli, injury, hypoxia-ischemia, and other adverse conditions affecting the normal function of the brain. Activated microglia take on an amoeboid morphology, proliferate, and migrate to the site of injury or damage where they protect the CNS from viruses and pathogens [31]. However, over activation and/or chronic activation of microglia with the excess production of inflammatory mediators may result in neurotoxic consequences [36]. Indeed, microglial cells can contribute to neurocognitive degeneration, as seen in the different forms of HAND, leading to an increase in proinflammatory chemokines and cytokines as well as neurotoxins which can affect both astrocytes and neurons and can cause neuronal damage [37].”
(page 14, paragraph 6 up to page 15 paragraph 2)
“5.2.4. miRNA and HIV Latency in HAND
Tha main role of miRNAs would be regulate the effects of virus propagation and replication, and so they are important agents in gene silencing and in post-transcriptional protein regulation. produced by viral genes involved in pathogenesis that interact with the host cell mRNAs [6, 121]. In vitro, HIV-1 blocks the expression of miRNA cluster miR-17/92 to efficiently replicate in peripheral blood mononuclear cells (PBMCs), monocytes and macrophages [122]. Of note, monocytes are less susceptible to HIV-1 infection than macrophages and so the presence of higher levels of anti-HIV miRNA (miRNA-382, miRNA-223, miRNA-150, and miRNA-28). In addition, elimination of these anti-HIV miRNAs in monocytes increases in HIV replication in these cells while the use of miRNA mimics in MDMs decreases HIV replication [123].
There are another original work that study the differentially expression in the brains of the miRNAs of persons living with HAND. These four miRNAs (miR-500a-5p, miR-34c-3p, miR-93-3p, and miR-381-3p) have been analyzed to regulate expression of the peroxisome biogenesis factors 2, 7,11B, 13, 19 (PEX2, PEX7, PEX11B, PEX13 and PEX19). Subsequent analyses indicated that elevated expression of these miRNAs was very common in HIV infection. These data can be demonstrated that the raised levels of mRNAs that downregulate peroxisomes can act as a process to change antiviral signaling that emanates from these organelles [124]. 125,126
There are numerous peroxisome-based diseases in humans including different serious diseases as Zellwegger syndrome spectrum disorders and rhizomelic chondrodysplasia punctata, leukodystrophy (inflammatory degeneration of white matter), similar to that observed in advanced HAND, termed HIV-associated dementia [124]. These data highlight the fact that even partially diminished function of peroxisomes can produce a severe neurological disease.
Activation of mitochondrial antiviral signaling (MAVS)-dependent signaling from peroxisomes by different RNA viruses produce activation of type III interferon. MAVS signaling from both peroxisomes and mitochondria is required for maximal anti-viral activity. Another pivotal process that involves PEX19 (crucial for degradation) was produced by flaviviruses in West Nile virus and Dengue virus, a critical peroxisome biogenesis factor. Flavivirus-infected cells contain significantly lower numbers of peroxisomes that reduce drastically type III interferon. Human cytomegalovirus HCMV protein vMIA vMIA interacts with peroxisomal MAVS and produces the fragmentation and disruption of peroxisomal morphology is not essential for this viral protein and require interaction with PEX19 to block antiviral signaling [124].
Another work have found three miRNAs to be elevated in both; brain tissue from rhesus macaques with and without simian immunodeficiency virus encephalitis (SIVE) and HIVE (miR-142-5p, miR-142-3p, and miR-21). [125] Further analysis revealed that miR-21 also stimulated N-methyl-D-aspartic acid receptors and targeted the mRNA of myocyte enhancer factor-2C (MEF2C). Similarly, Noorbakhsh et al. (2010) identified the differential expression of multiple miRNAs in frontal lobe white matter by comparing HIV-negative and HIVE cases matched by age and sex [126], using the standard two-fold cut-off as the threshold for further analysis in their expression profiling study. Thus, given all these results, strategic miRNA manipulation could provide a potent therapeutic tool against HAND.
Latent HIV-1 reservoirs represent the largest roadblock to the complete eradication of HIV-1 from infected individuals. However, according to the ‘kick and kill’ strategy, viruses can be activated in these reservoirs using a range of latency-reversing agents, including histone deacetylase inhibitors (HDACi), histone methyltransferase inhibitors (HMTi), and DNA methyltransferase inhibitors (DNMTi), protein kinase C (PKC) agonists, and several other small molecules.”
(page 16; paragraph 3 up to page 17 paragraph 3)
“6.2. Immune-based strategies
The use of cytokines and other agents that revert negative regulation of immune activation are also considered good immune-based therapies because of their capacity to both antagonize virus silencing and restore immune functions. It has not yet been established whether reactivated cells are killed by cytopathic effects or are recognized and eliminated by the immune system [132]. Cytokines including IL-2, IL-7, and IL-15 with putative roles in HIV-1 control are currently being tested in humans. However, further studies will be required to elucidate the effective potential of these cytokines as candidates to deplete the HIV reservoir. In addition, preventative HIV vaccine trials have generally been ineffective, with only one clinical study so far having shown a protective effect, although there was no noticeable effect on the degree of viremia or CD4+ T cell numbers in patients who were eventually infected [133].
Furthermore, there is evidence that reversal of HIV latency may not necessarily lead to clearance of reactivated cells, with several therapeutic vaccine strategies having been tested in this context [134]. Therapeutic vaccination could re-stimulate CTL (CD4+ T cells) responses, mimicking the situation of patients in which viral replication is controlled without treatment and without progression to AIDS [135]. This method may be useful in reactivating the virus because the HIV-1 env and pol antigens were reported to activate most of CD8+ T cells harboring proviral DNA, inducing HIV-1 replication [135]. In the NCT00107549 clinical trial, patients were immunized with a poxvirus vaccine engineered to express HIV-1 antigens; a significant, albeit transient, decrease in replication-competent HIV-1 in the resting T-cell reservoir was observed in these patients [136].
Some vaccination treatments and pathogen infections have been found to transiently increase viral RNA in the plasma of HIV-1-infected patients during ART. In fact, the use of TLR agonists have recently been suggested to both reactivate HIV-1 from latently infected cells and to boost HIV-specific cytotoxic CD8+ T cell immunity [137]. On the contrary, the use of TLR antagonists such as chloroquine is thought to lower immune activation. Treatments with hydroxychloroquine or chloroquine have been evaluated in the clinical setting in ART-treated [138] and ART-naïve patients [139] and was reported to reduce immune activation, albeit with little or no effect on CD4+ T cell recovery. However, in a randomized trial in naïve, non-progressing patients, hydroxychloroquine treatment resulted in the worsening of CD4+ T cell loss [139].
Strategies aimed at the recovery of immune system functionality are pivotal in current studies. An important immunological target is programmed cell death protein 1 (PD-1), a receptor known for its role in immune exhaustion. PD-1 receptor is upregulated in HIV-specific and non-specific CD4+ and CD8+ T cells, limiting their functions [140], and moreover, the compartment of memory T cells expressing elevated levels of PD-1 contain more proviral HIV-1 DNA than PD-1 low cells [140]. In patients on cART, a consistent association between the frequency of PD-1-expressing cells and the size of the reservoir has been shown [141]. Indeed, PD-1 is expressed at lower levels in ‘elite controllers’ and ‘long-term non-progressors’ (LTNPs) compared to typical progressors [142]. These recent data indicate that PD-1 high cells form a crucial reservoir for the virus and that PD-1 inhibition may also support beneficial effects by reducing the latent reservoir.
Some work has also tried to test whether blocking this negative regulator of immune activation may both activate HIV-1 transcription and reverse immune exhaustion by relieving a functional block on virus-specific CD8 memory T cells [143]. Indeed, in vivo studies testing anti-PD1 antibodies on non-human primate models of HIV infection have shown positive effects on immune system restoration. These findings improved immune responses in the blood and gut and were associated with significant reductions in plasma viral load, greater antibody responses to both SIV and non-SIV antigens, and prolonged survival of SIV-infected macaques [144]. Furthermore, PD-1 blockade reduced persistent immune activation and showed evidence of the restoration of mucosal barrier integrity and bacterial translocation [144]. A study by an AIDS clinical trial group (ACTG 5301) evaluating the safety and efficacy of PD-1 blockade to reduce the latent HIV reservoir in patients in cART is currently being developed.
The antifibrotic effects of angiotensin-converting enzyme inhibitors and angiotensin receptor blockers through suppression of transforming growth factor-β in different clinical settings has more recently has been studied in the context of HIV infection, with clinical trials is currently in phases I and II (NCT01535235; ACTG 5317). Their capacity to improve HIV-specific immune responses and to reduce the viral reservoir in lymphoid tissues is also being assessed. Two other promising anti-inflammatory molecules are the peroxisome proliferator-activated receptor agonists pioglitazone and leflunomide, which are both well tolerated and effective in the treatment of chronic inflammation and reduce the metabolic syndromes associated with prolonged ART [145].
6.3. Stem cell transplantation
An important case report by Allers et al. demonstrated the transplant of homozygous CCR5 delta32 stem cells into an HIV-1 infected patient that suffers acute myeloid leukemia, resulting in a solution for the disease whereby the ‘Berlin patient’ was able to stop antiretroviral treatment with viremic rebound [146] (Figure 4). No virus was identified in multiple samples of non-CD4 T cell populations from this patient when using different assessment techniques [147]. Nevertheless, it remains undetermined whether this stem cell transplantation strategy was able to overcome the HIV reservoir in macrophages and microglia. Of note, the prior administration of gemtuzumab ozogamicin (an anti-CD33 monoclonal antibody) in this patient may have targeted myeloid cells for depletion (Figure 4).”
Referenced added in the bibliography
[30] Ginhoux F., Guilliams M. (2016) Tissue-Resident Macrophage Ontogeny and Homeostasis. Immunity. 15;44(3):439-449. doi: 10.1016/j.immuni.2016.02.024.
[31] Prinz M., Jung S., Priller J. (2019) Microglia Biology: One Century of Evolving Concepts. Cell. 3;179(2):292-311. doi: 10.1016/j.cell.2019.08.053.
[36] Kaur C., Sivakumar V., Zou Z., Ling E.A. (2014) Microglia-derived proinflammatory cytokines tumor necrosis factor-alpha and interleukin-1beta induce Purkinje neuronal apoptosis via their receptors in hypoxic neonatal rat brain. Brain Struct Funct. 219(1):151-70. doi: 10.1007/s00429-012-0491-5.
[124] Xu Z., Asahchop E.L., Branton W.G., Gelman B.B., Power C., Hobman T.C. (2017) MicroRNAs upregulated during HIV infection target peroxisome biogenesis factors: Implications for virus biology, disease mechanisms and neuropathology. PLoS Pathog. 13(6):e1006360. Published 2017 Jun 8. doi:10.1371/journal.ppat.1006360
[133] Jilg N., Li J.Z. (2019) On the Road to a HIV Cure: Moving Beyond Berlin and London. Infect Dis Clin North Am. 33(3):857-868. doi:10.1016/j.idc.2019.04.007
[134] Deng K., Pertea M., Rongvaux A., Wang L., Durand C.M., Ghiaur G., et al. (2015) Broad CTL response is required to clear latent HIV-1 due to dominance of escape mutations. Nature.517(7534):381-5. doi: 10.1038/nature14053.
[135] García F, León A, Gatell JM, Plana M, Gallart T. Therapeutic vaccines against HIV infection. Hum Vaccin Immunother. 2012 May;8(5):569-81. doi: 10.4161/hv.19555.
[136] Persaud D., Luzuriaga K., Ziemniak C., Muresan P., Greenough T., Fenton T., et al. (2011) Effect of therapeutic HIV recombinant poxvirus vaccines on the size of the resting CD4+ T-cell latent HIV reservoir. AIDS. 25(18):2227-34. doi: 10.1097/QAD.0b013e32834cdaba.
[137] Novis C.L., Archin N.M., Buzon M.J., Verdin E., Round J.L., Lichterfeld M., et al. (2013) Reactivation of latent HIV-1 in central memory CD4⁺ T cells through TLR-1/2 stimulation. Retrovirology. 24;10:119. doi: 10.1186/1742-4690-10-119.
[138] Ahlenstiel C. L., Symonds G., Kent S. J., Kelleher A. D. (2020) Block and Lock HIV Cure Strategies to Control the Latent Reservoir. Frontiers in cellular and infection microbiology, 10, 424. https://doi.org/10.3389/fcimb.2020.00424
[139] Pang K. M., Castanotto D., Li H., Scherer L., Rossi J. J. (2018) Incorporation of aptamers in the terminal loop of shRNAs yields aneffective and novel combinatorial targeting strategy. Nucleic Acids Res.46:e6.doi: 10.1093/nar/gkx980[142]
[140] Battistini A., Sgarbanti M. (2014) HIV-1 latency: an update of molecular mechanisms and therapeutic strategies. Viruses. Apr 14;6(4):1715-58. doi: 10.3390/v6041715.
[141] Hatano H., Jain V., Hunt P.W., Lee T.H., Sinclair E., Do T.D., et al. (2013) Cell-based measures of viral persistence are associated with immune activation and programmed cell death protein 1 (PD-1)-expressing CD4+ T cells. J Infect Dis. 208(1):50-6. doi: 10.1093/infdis/jis630. Epub 2012 Oct 22.
[142] Kulpa D.A., Lawani M., Cooper A., Peretz Y., Ahlers J., Sékaly R.P. (2013) PD-1 coinhibitory signals: the link between pathogenesis and protection. Semin Immunol. 25(3):219-27. doi: 10.1016/j.smim.2013.02.002.
[143] Titanji K., Velu V., Chennareddi L., Vijay-Kumar M., Gewirtz A.T., Freeman G.J., et al. (2010) Acute depletion of activated memory B cells involves the PD-1 pathway in rapidly progressing SIV-infected macaques. J Clin Invest. 120(11):3878-90. doi: 10.1172/JCI43271.
[144] Dyavar Shetty R., Velu V., Titanji K., Bosinger S.E., Freeman G.J., Silvestri G., et al. (2012). PD-1 blockade during chronic SIV infection reduces hyperimmune activation and microbial translocation in rhesus macaques. J Clin Invest. 122(5):1712-6. doi: 10.1172/JCI60612.
[145] Mencarelli A., Francisci D., Renga B., D'Amore C., Cipriani S., Basile F., et al. (2012) Ritonavir-induced lipoatrophy and dyslipidaemia is reversed by the anti-inflammatory drug leflunomide in a PPAR-γ-dependent manner. Antivir Ther. 17(4):669-78. doi: 10.3851/IMP2039.
